# The *Toxoplasma* micropore mediates endocytosis for selective nutrient salvage from host cell compartments

Wenyan Wan [1,6], Hui Dong[1,6], De-Hua Lai [2,6], Jiong Yang[2], Kai He [1], Xiaoyan Tang[1], Qun Liu[1], Geoff Hide [3], Xing-Quan Zhu[4], L. David Sibley[5], Zhao-Rong Lun[2] & Shaojun Long [1] ✉

Apicomplexan parasite growth and replication relies on nutrient acquisition from host cells, in which intracellular multiplication occurs, yet the mechanisms that underlie the nutrient salvage remain elusive. Numerous ultrastructural studies have documented a plasma membrane invagination with a dense neck, termed the micropore, on the surface of intracellular parasites. However, the function of this structure remains unknown. Here we validate the micropore as an essential organelle for endocytosis of nutrients from the host cell cytosol and Golgi in the model apicomplexan *Toxoplasma gondii*. Detailed analyses demonstrated that Kelch13 is localized at the dense neck of the organelle and functions as a protein hub at the micropore for endocytic uptake. Intriguingly, maximal activity of the micropore requires the ceramide de novo synthesis pathway in the parasite. Thus, this study provides insights into the machinery underlying acquisition of host cell-derived nutrients by apicomplexan parasites that are otherwise sequestered from host cell compartments.

The phylum Apicomplexa comprises thousands of intracellular parasites that occupy divergent niches[1]. *Plasmodium falciparum*, the agent responsible for human malaria, proliferates in erythrocytes and hepatocytes. *Cryptosporidium*, the cause of acute gastrointestinal diseases, multiplies in epithelial cells. *Toxoplasma gondii*, the most successful zoonotic parasite, replicates in almost all nucleated cells of warm-blooded vertebrate hosts. These parasites reside in host cells with diversified membranous compartments at some, or all, stages of their life cycles, and acquire nutrients from these compartments. *T. gondii* establishes a replication permissive niche in cells, within the parasitophorous vacuole (PV), which is segregated from most host endo-membranous systems required for cellular function[2]. This unique niche poses complex challenges for the parasite to scavenge nutrients to support its own growth.

Previous studies have revealed several mechanisms that are exploited to internalize host nutrients and organelles to the parasitophorous vacuole by crossing the PV membrane (PVM). These include a proteinaceous pore[3], the Endosomal Sorting Complex Required for Transport (ESCRT)[4] and the Host Organelle–Sequestering Tubulostructures (H.O.S.T.)[5,6]. The PV may act as a digestive compartment to process substrates prior to translocation across the parasite plasma membrane (PPM)[5,7,8].

Our current understanding of the *T. gondii* PPM in nutrient acquisition stems from investigations into PPM transporters. These

[1]National Key Laboratory of Veterinary Public Health Security and School of Veterinary Medicine, China Agricultural University, 100193 Beijing, China. [2]MOE Key Laboratory of Gene Function and Regulation, State Key Laboratory of Biocontrol, School of Life Sciences, Sun Yat-Sen University, Guangzhou 510275, China. [3]Biomedical Research and Innovation Centre and Environmental Research and Innovation Centre, School of Science, Engineering and Environment, University of Salford, Salford M5 4WT, UK. [4]College of Veterinary Medicine, Shanxi Agricultural University, Taigu 030801 Shanxi, China. [5]Department of Molecular Microbiology, Washington University School of Medicine in Saint Louis, Saint Louis, MO 63110-1093, USA. [6]These authors contributed equally: Wenyan Wan, Hui Dong, De-Hua Lai. ✉e-mail: LongS2018@163.com

transporters are responsible for transporting adenosine[9,10], glucose[11], lactate/pyruvate[12], tyrosine[13], arginine and lysine[14] and folate[15]. Despite this, studies have revealed that the parasite also scavenges host proteins[16], fatty acids[17,18], myo-inositol[19], serine, ethanolamine, choline[20], ceramide, sphingolipids[6,21], cholesterol[5], vitamins and co-factors[22,23], from the host cell. However, there is a huge gap in our understanding of the mechanisms of acquisition of these and other nutrients by *T. gondii*.

Endocytosis is a fundamental mechanism for both nutrient acquisition and membrane homeostasis. Clathrin mediated endocytosis (CME) is recognized as the most studied mechanism[24]. However, underneath the PPM, parasites have an inner membrane complex (IMC) that may protect against invagination of the PPM and prevent endocytosis. Ultrastructural studies have repeatedly reported a PPM invagination structure, termed the micropore, on the surface of intracellular parasites, and this invagination is unique as it crosses the IMC[25,26]. The micropore was first observed in the sporozoites of avian malaria parasites[27] and in the cysts (bradyzoites) of *T. gondii*[28]. It was later found in almost every stage of apicomplexan parasites, when studied by electron microscopy[25,29], and in related parasites such as parasitic dinoflagellates, protococcidians and colpodellids[30–32]. The micropore usually appears as an invagination of the PPM with laterally thickened material around the neck[25,26]. Intriguingly, in the erythrocytic forms of malaria parasites, a structure appears to have invaginations in both the PPM and the PVM. This has been termed the cytostome and is considered potentially to be a specialized version of the micropore used for hemoglobin uptake[33–36]. However, the endocytic processes in most apicomplexan parasites are not as obvious as that of hemoglobin uptake in *Plasmodium*[37]. Only a minority of extracellular parasites have actually been observed to take up extracellular tracers[26,38]. Yet, there is lack of solid evidence for the cytostome being the micropore and the micropore acting as a site for endocytosis and/or as a functional structure in related parasites. In addition, studies suggesting other potential sites of endocytosis in *T. gondii* have further increased the mystery surrounding endocytosis and nutrient acquisition[6,39]. Nevertheless, *T. gondii* could serve as a good model organism to dissect out which proteins define the endocytic processes and how they operate. Such knowledge would improve our understanding of how *T. gondii* and other apicomplexan parasites scavenge nutrients from host cells.

In this study, we show that the micropore buds an endocytic vesicle at its base, suggesting that it has a function in active endocytosis in *T. gondii*. Using comparative knowledge about classical proteins (e.g., EPS15 and AP2) involved in clathrin mediated endocytosis (CME) and a proximity labeling approach in *T. gondii*[40], we identified the proteome and established a protein network−the endocytosome where Kelch13 functions as a protein hub. Our study clarifies the essential role of the micropore in *T. gondii* and provides the first detailed insights into the elusive mechanism driving the acquisition of nutrients from complex host cellular compartments in apicomplexan parasites.

## Results

### Discovery of proteins localized to the micropore in *T. gondii*

Classical CME proteins, such as clathrin, dynamin and EpsL, have been implicated in organelle biogenesis in *T. gondii*[41–45]. To understand the potential mechanisms of endocytosis in *T. gondii*, we postulated that other CME proteins might be utilized in this organism. To identify these proteins, we searched the ToxoDB database (https://toxodb.org/toxo/app/) for proteins containing EH (EPS15 Homology)-, BAR (Bin-Amphiphsin-Rvs)-domains and AP2 (adaptin, AP) homologs, based on their involvement in endocytosis in mammals and yeast[24]. We identified nine proteins (Supplementary Fig. 1a), namely two EPS15-like proteins, two BAR domain proteins and five possible AP2 subunits including AP2αc and AP2αn that are annotated as such in the ToxoDB

database. To localize these proteins, we generated protein-6Ty fusion lines using CRISPR/Cas9[46] (Supplementary Data 1–3). Immunofluorescence assays (IFA) showed that the EPS15 and AP2 subunits, excepting AP2β, were localized to similar foci at the parasite surface (Supplementary Fig. 1b). AP2β may appear to have retained the binding ability to clathrin and is localized at the Golgi, while the other AP2 subunits, highly divergent in the Apicomplexa[47], are not. To clarify any possible co-localization, 6HA epitope tags were added to the AP2 components in the EPS15-6Ty line, creating two fusion proteins in the same lines. Confocal imaging revealed that AP2αc, AP2αn, AP2μ and AP2σ were highly co-localized with EPS15 (Fig. 1a). Based on this localization feature, we postulated that these proteins are localized at the micropore.

We then attempted to identify the proteome of the foci using a permissive biotinylation approach, which detects proteins that are proximal to a bait protein (e.g., EPS15) fused with a highly efficient biotin ligase TurboID[48]. Biotin labeling detected additional protein bands on blots, compared to the parental line. As shown by staining using labeled streptavidin, these proteins were concentrated at the foci of the potential micropore and also a weak additional focus was identified in the apicoplast (Supplementary Fig. 1c). Replicate mass-spectrometry (MS) datasets were used to create a volcano plot comparing the TurboID fusion to the parental line (Supplementary Data 4 and Supplementary Fig. 1d). To identify new proteins, we selected ten significant candidates and four candidates with *p* values close to 0.05. CRISPR-Cas9 was then used to tag candidate proteins with 6Ty (Supplementary Data 5). Five new candidates that localized to the foci were identified (Supplementary Fig. 1e). Confocal imaging with 6HA fusions to the new candidates in the EPS15-6Ty line showed clear co-localization with EPS15 (Fig. 1b). Finally, we identified ten proteins, among which Kelch13, UBP1, EPS15, AP2μ, AP2αc (PfKIC4) and PPG1 (PfKIC2) are shared with proteins identified at the cytostome in *P. falciparum* reported in a recent study[49]. Moreover, two lineage specific proteins (i.e., MPP1 and MPP2, MicroPore Proteins) and another two AP2 subunits (i.e., AP2an and AP2σ) were identified at the *Toxoplasma* micropore, while more additional proteins were identified at the cytostome of *P. falciparum* (Supplementary Table 1).

To precisely localize the proteins, we utilized a higher resolution approach - stochastic optical reconstruction microscopy (STORM) to examine the distribution pattern of Kelch13 in these parasites. Kelch13 was chosen as it is the most conserved protein of the group, as discussed below. STORM revealed that Kelch13 (K13) was mainly concentrated in a narrow ring with a diameter ~200 nm embedded within the IMC (Fig. 1c). Consistent with this pattern, serial immunoEM revealed that Kelch13, labeled with gold particles, was concentrated in a location adjacent to a cup-like structure on continuous sections (100 nm thick) (Fig. 1d). A clear preparation of serial TEM sections revealed that the cup-like structure is the micropore with an invagination of the PPM that is encircled by an electron dense ring at the neck (Fig. 1e). Furthermore, the PPM is coated with a less electron dense region at the base. The dimensions of dense ring are ~172 nm in diameter, ~50 nm in thickness and ~67 nm in length (*n* = 24) (Fig. 1f). To gain a three-dimensional view, TEM tomography reconstruction illustrated that a PVM conduit was not observed inside the micropore, however a budding endocytic vesicle at the base was observed. This budding endocytic vesicle had a single membrane that enclosed a material that was similar to the PV lumen (Fig. 1g and Supplementary Movie 1). Overall, we have demonstrated that one representative of the ten proteins, Kelch13, is localized to the micropore in *T. gondii*, and the micropore has functional budding endocytic vesicles.

### Kelch13 mediates a key interacting complex at the micropore

As predicted, the above studied micropore proteins contain various domains that include an EH domain, Kelch repeats, a BTB domain, AP complex domains and a microtubule binding domain (Supplementary

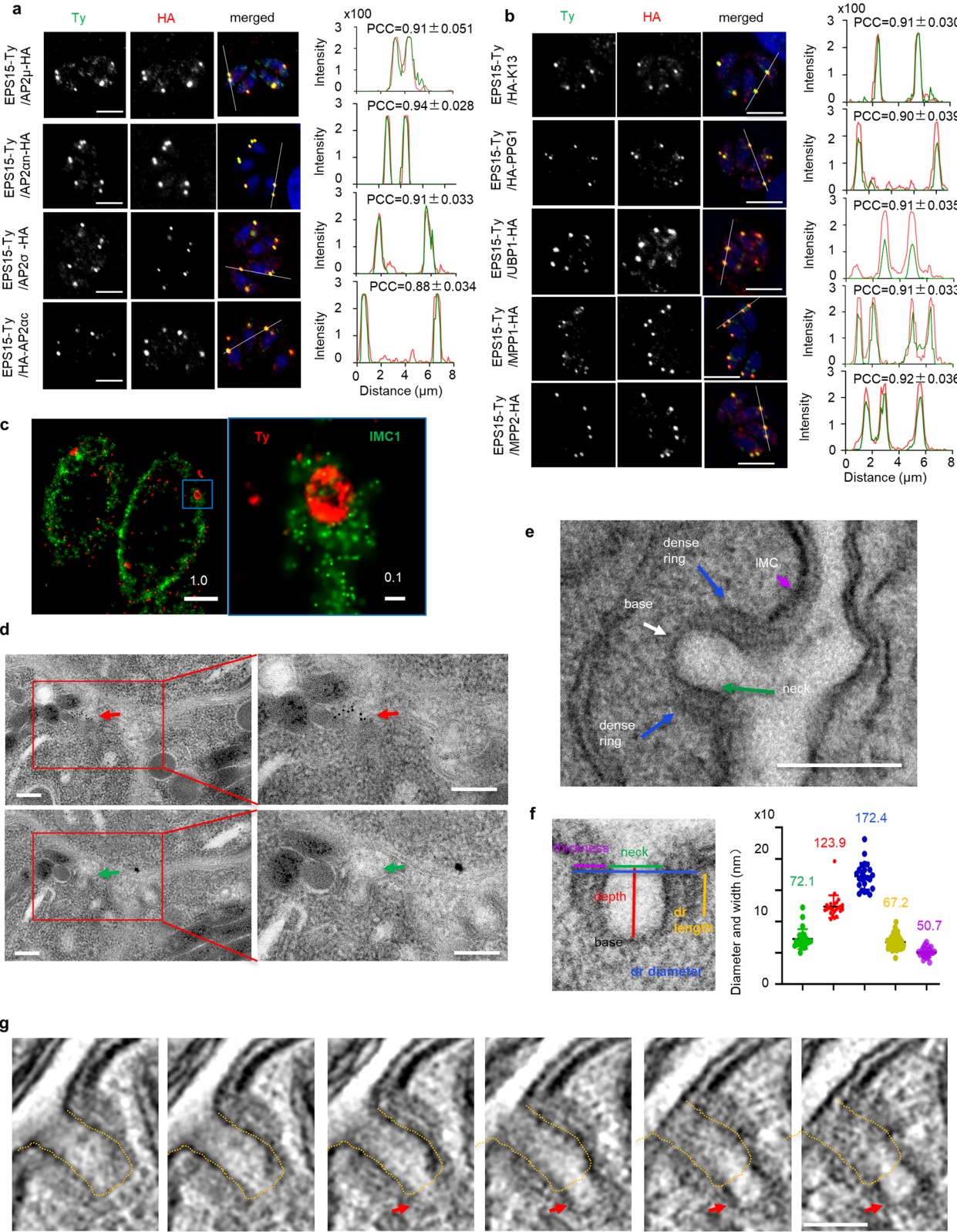

Fig. 2a, b). It is noteworthy that both EH domains and Kelch/BTB domains mediate protein-protein interactions[50,51]. In addition, many of these proteins contain intrinsic disordered regions (IDRs) and coiled coils (CC), as predicted by InterPro and a newly developed webserver flDPnn[52] (Supplementary Fig. 2b, c). These domains are often associated with having multiple binding partners and adopting multiple conformations[53,54], indicating the potential existence of a protein

interaction network. Analysis of evolutionary conservation showed that Kelch13 is the most conserved of these proteins amongst the apicomplexans, although the gregarines have a shortened polypeptide that retains only the BTB domain (Fig. 2a, Supplementary Fig. 2d and Supplementary Data 6). Other proteins have different degrees of conservation within the phylum (Fig. 2a and Supplementary Data 6). Moreover, many of the genes, such as Kelch13, UBP1, AP2αc, EPS15,

**Fig. 1 | The discovery of proteins localized to the micropore in *T. gondii*.**
**a, b** Confocal co-localization of protein fusions with EPS15-6Ty, showing confocal imaging of EPS15-6Ty (green) and 6HA-fusions (red). The Pearson correlation coefficient (PCC) was analyzed with mean ± SEM using the intensities over the white line by NIS software. Three independent experiments were performed with similar outcomes, and The PCC shown were the averages and standard deviation ($N = 6$). Scale = 5 μm. **c** Stochastic optical reconstruction microscopy (STORM) imaging of Ty-Kelch13. Ty-Kelch13 was stained with Ty (red) and IMC1 (green was the control). The blue square enlarged as indicated by the relative lengths of the scale bars (-6x). **d** Serial immuno-electron microscopy (immunoEM) of Ty-Kelch13, showing deposition of gold particles (red arrowheads) at a position adjacent to the micropore cup (green arrowheads) on two continuous sections. Red squares enlarged. Scale = 200 nm. **e** Ultra-structure of the micropore, showing a cup-like structure

with a neck (green arrow), a base (white arrow) and electron-dense ring (deep blue arrow). The IMC, pink arrowhead. Scale = 200 nm. **f** Size measurement of the micropore, showing the measurements taken: neck diameter (green), cup depth (red), thickness (pink), diameter (deep blue) and depth (yellow) of the dense ring. The sizes were plotted in the same colors, and the data were shown ($N = 24$) with mean ± SEM. More than three independent experiments were performed on the TEM. **g** Serial images of a reconstructed tomography analysis of a micropore with a budding endocytic vesicle (red arrowheads) at the base. The yellow dotted line was drawn to indicate the plasma membrane of the micropore on the first image at the left, and it was placed on other images to show the budding endocytic vesicle. The single images were next to each other at 9.5 nm. Scale = 200 nm. Three independent experiments were performed with similar outcomes and one representative of the results was shown.

have low CRISPR fitness scores[55] (Fig. 2a), indicating these genes are crucial for parasite growth. These features attracted us to map the protein interactions, using a proximity-based approach, as previously reported for the conoid interactome[40]. We used TurboID fusions tagged to Kelch13, PPG1, UBP1, MPP1, MPP2, AP2μ (AP2μ as the representative of AP2). Analysis of biotinylation activity effectively showed labeling of the micropore, as well as the apicoplast (Supplementary Fig. 2e). The resulting MS datasets contained many peptides belonging to the IMC proteins and microtubules (Fig. 2b and Supplementary Data 7), indicating the likely proximity of PPG1, AP2μ and MPP1 to the inner membrane complex (IMC) and microtubules.

To identify potential interactions from the complex datasets, we used an R-based program Straightforward Filtering IndeX program (SFINX), as previously described[40]. SFINX provides an interactome and statistical criteria for meaningful interactions[56]. Here we included the full interactome network with a strictness = 4 (*p* value < 0.00005) (Fig. 2c and Supplementary Movie 2, Supplementary Data 8 contains the input file for SFINX and Supplementary Data 9 contains a summary of the output). The local SFINX run revealed both a core interactome and other branches (Fig. 2c). Importantly, all the partners of Kelch13 were also identified by EPS15-TurboID and TurboID-PPG1. This result suggested that Kelch13 forms a prominent node in the core interactome that closely interacts with an overlapping set of proteins shared with EPS15 and PPG1 (Fig. 2c and Supplementary Data 9). Intriguingly, AP2αc is one of the interacting proteins at the core interactome, while the other AP2 adaptors are only seen within the branches (Supplementary Data 9), indicating that AP2αc is different from the others. Additionally, dynamin C (DrpC) is also one of the interacting proteins associated with the core nodes. DrpC plays a role in mitochondrial fission in dividing parasites[43,57]. Our confocal imaging using the DrpC-6HA/EPS15-6Ty fusions confirmed the co-localization of DrpC with EPS15 in mature parasites (Supplementary Fig. 2f). This localization is consistent with the previous co-immunoprecipitation observations with DrpC, which intriguingly detected Kelch13 and AP2 subunits[43], supporting reliability of our proximity-based protein interaction network. In summary, Kelch13 may serve as a protein hub, which mediates a protein network (termed the endocytosome) at the micropore.

**The micropore proteins are essential for parasite growth in vitro**
To investigate functions of the micropore, we used the auxin-inducible degron (AID) system for conditional knockdown of proteins in *T. gondii*[46,58]. We added the AID-HA or AID-Ty fusion to the N-terminus or C-terminus of the micropore proteins in the parental line TIR1 using a CRISPR/Cas9 approach (Supplementary Fig. 3a, b). We successfully created ten single AID fusion lines. We did not include DrpC as preliminary studies revealed that it was unable to be degraded by the system. To study the core interactome, we added the Ty- and myc-AID to Kelch13 and PPG1 at their N-termini, respectively, in the EPS15-AID-HA line. This approach created a double AID fusion line EPS15-AID-HA/Ty-AID-Kelch13 (dKD) and a

triple AID fusion line EPS15-AID-HA/Ty-AID-Kelch13/myc-AID-PPG1 (triple KnockDown, tKD). The correct fusions at specific genes were confirmed by diagnostic PCRs (Supplementary Fig. 3c, d). The AID fusions appeared in foci in IMC gaps congruent with a correct location and the addition of auxin (IAA) resulted in their efficient degradation to almost undetectable levels (Supplementary Fig. 4). However, the parasite morphology did not show any obvious changes following 24 h of auxin induction (Supplementary Fig. 4). To test whether the AID fusions were essential for parasite growth, we examined the ability of parasites to form plaques on host cell monolayers. Significantly, when grown in the presence of auxin, the AID-kelch13, double and triple AID cell lines formed no apparent plaques, and the AID fusions of AP2αc, EPS15, MPP1, UBP1, AP2αn and AP2σ had significantly fewer and smaller plaques (Fig. 2d–f and Supplementary Fig. 5a, b). This outcome roughly matched the low CRISPR fitness scores for the genes. Detailed examinations of parasite replication status revealed the contribution of replication defects to the fitness of the AID lines mentioned above (Fig. 2g and Supplementary Fig. 5c). These results consistently support the notion that Kelch13 functions as a protein hub and contributes to parasite replication in combination with other micropore proteins.

**The Kelch13 complex controls selective salvage of host materials**
To investigate the potential role of the micropore proteins in nutrient salvage, we first assessed uptake of host cell cytosolic GFP that is expressed by a Lenti-viral expression system II. The fitness-conferring AID lines and the parental line TIR1 were further genetically modified to delete the *cpl* gene (Δ*cpl*) that encodes a protease CPL capable of digesting host cell-derived GFP in the vacuolar compartment (VAC)[16]. By growth of these Δ*cpl* lines in the GFP-expressing host cells in auxin for 36 h, we observed that the percentage of parasites containing host cell-derived GFP dramatically dropped by comparison with the TIR1/Δ*cpl*, with the exception of MPP1-AID/Δ*cpl* (Fig. 3a and Supplementary Fig. 6a). As the TIR1-AID system usually down-regulates proteins very quickly, we then wondered if down-regulation of the key micropore protein K13 (AID-K13) would cause a rapid drop in proportion of parasites containing GFP. Western blot analysis showed that AID-K13 dropped to undetectable level at 2 h and kept no changes at the following hours (Fig. 3b, c). To test GFP acquisition, we added the CPL inhibitor LHVS (10 μM) to the culture, while the auxin induction was initiated, as illustrated in Fig. 3d. This strategy greatly reduced initial accumulation of GFP in parasites prior to the induction, allowing examination of the direct impact on GFP acquisition upon depletion of Kelch13. Using this approach, we showed that depletion of K13 significantly decreased GFP acquisition in the early induction time points (2, 4 and 6 h of induction) (Fig. 3e). These results suggested that Kelch13 is involved in the salvage of host cytosolic proteins together with other micropore proteins. Therefore, in the following part of the study, we mainly focused on the key AID (fusion) lines of EPS15-AID, AID-Kelch13 and tKD.

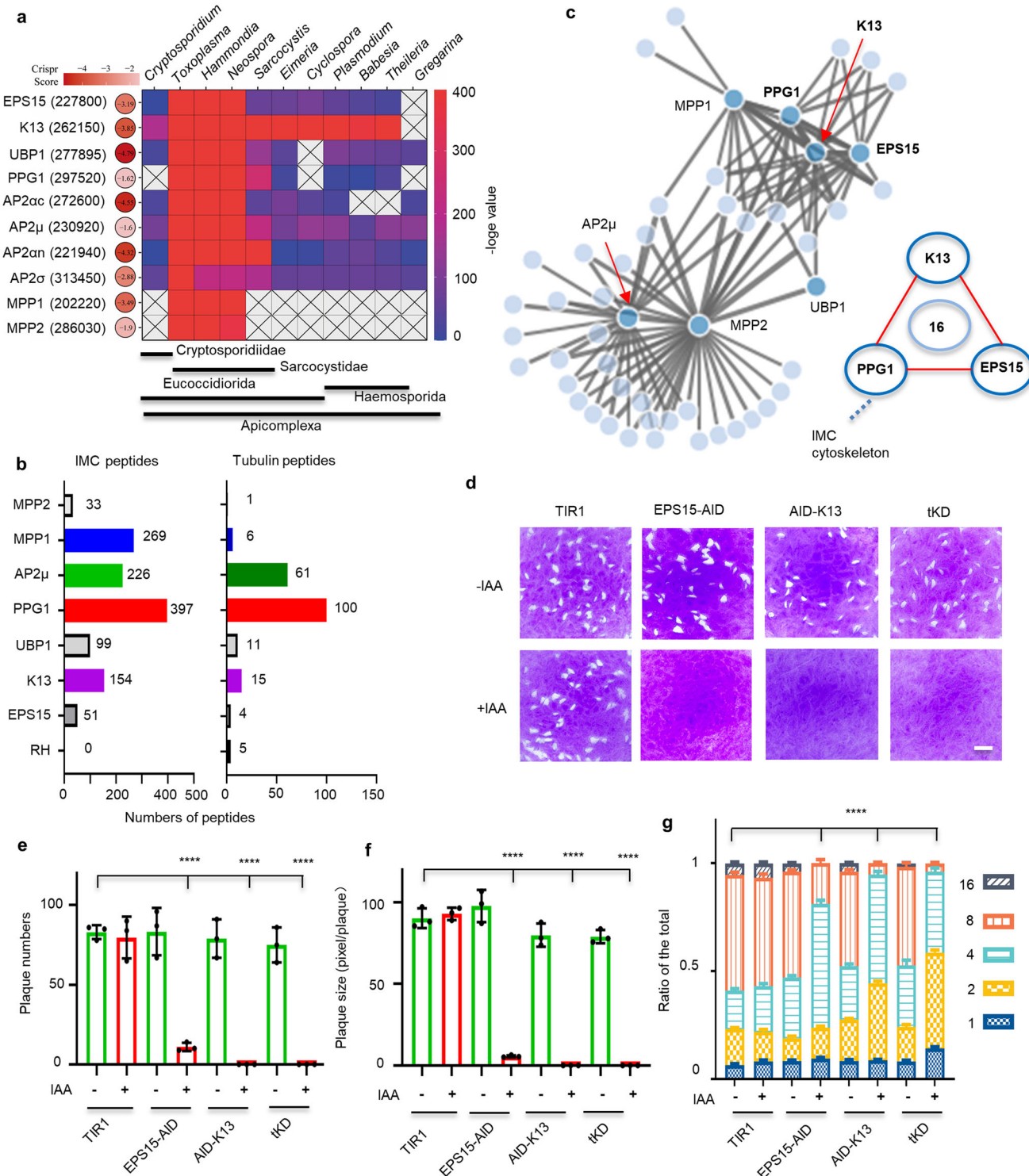

**Fig. 2 | Kelch13 mediates the protein interaction network at the micropore.**
**a** Conservation of micropore proteins in representative species of the apicomplexans. The protein names with the corresponding ID (in brackets) were listed, followed by CRISPR fitness scores and the heatmap[55] by the side of a conservation heatmap. The heatmaps were correspondingly scaled by $-\log E$ values and CRISPR scores. **b** Peptide numbers of IMC proteins and microtubules identified by proximity biotin labeling. **c** The proximity-based interactome of micropore proteins, showing interactions of the bait (deep blue) and the prey (light blue) filtered through a R-based SFINX package for 7 TurboID fusions ($N = 2$ datasets for each) with strictness = 4 ($p$ value < 0.00005). A hypergeometric test was used in the SFINX analysis. The core nodes K13, PPG1 and EPS15 shared the same set of interactors. **d**–**f** Plaque formation of AID lines grown in ±IAA for 7 days. EPS15, PPG1 and K13 were separately fused with AID in tKD. Plaque numbers and sizes were scored and plotted (**e**, **f**). The sizes were measured for 18 plaques in each independent experiment. Scale = 0.5 cm. **g** Parasite replication for ±IAA parasites for 24 h, showing ratios of different parasites/vacuole in the total ($n > 100$ for each replicate). In comparing EPS15-AID +IAA vs. TIR1 −IAA, $p < 0.0001$ for 4 and 8 parasites/vacuole and $p < 0.01$ for 2 parasites/vacuole; in comparing AID-K13+ IAA vs. TIR1 −IAA and tKD +IAA vs. TIR1 −IAA, $p < 0.0001$ for 2, 4 and 8 parasites/vacuole. Auxin treatment removed all the 16 parasites/vacuole for the AID lines. **e**–**g** Mean ± SD for three independent experiments with triplicates, analyzed by one-way or two-way ANOVA with Tukey's multiple comparison, ****$p < 0.0001$.

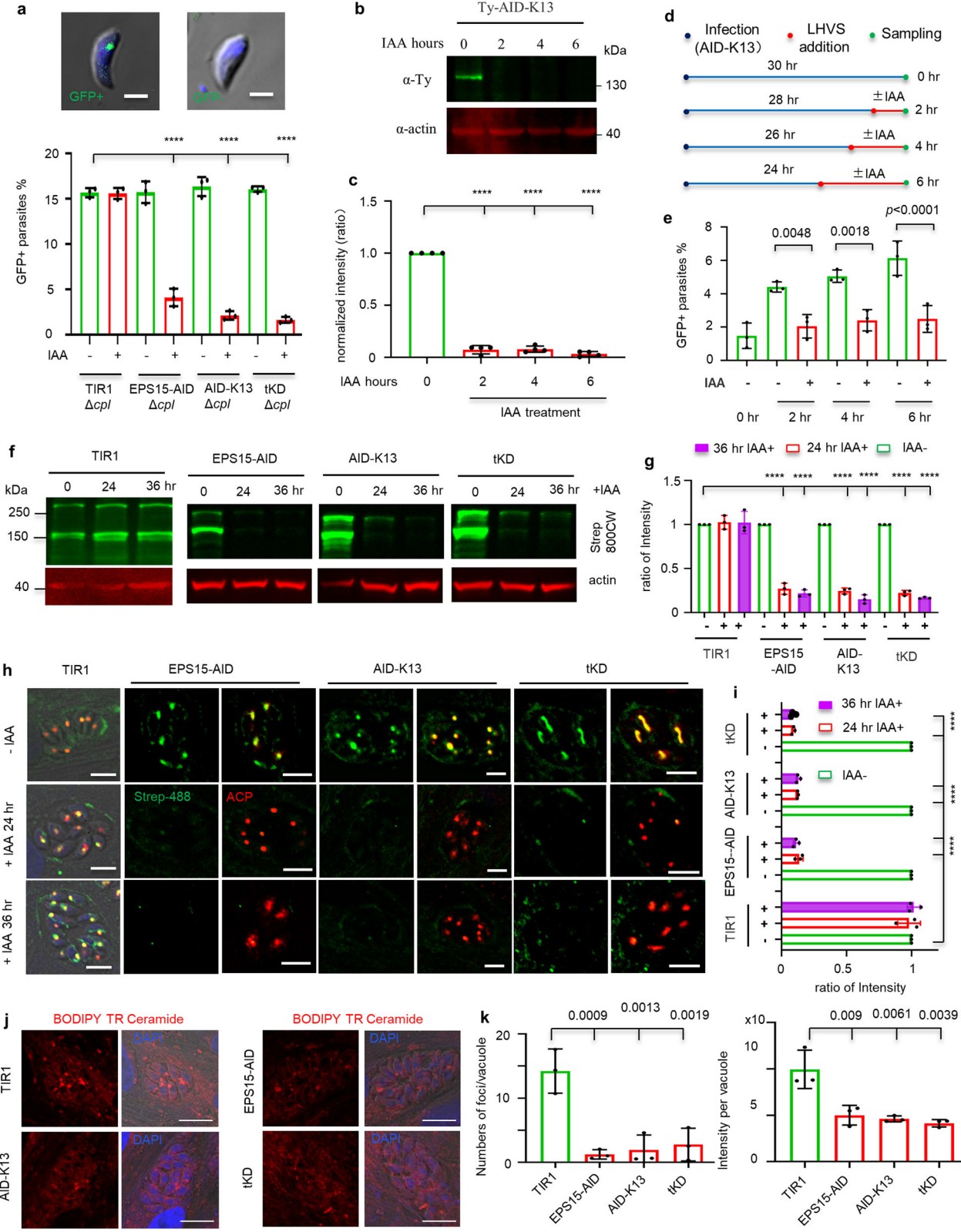

We subsequently decided to study the potential salvage of metabolites and co-factors from different host compartments such as the cytosol (biotin), the Golgi apparatus (ceramide)[6,21], lysosome (cholesterol)[5], lipid droplets[59,60] and the mitochondrion (fatty acids and lipoic acid)[17,23]. Biotin is a water-soluble vitamin and is important in minute quantities for the functioning of three carboxylases including ACC1 (288 kDa) in the apicoplast, ACC2 (365 kDa) in the cytosol and pyruvate carboxylase (PC) (152 kDa) in the mitochondrion of *T. gondii*[61,62]. We used the strong affinity binding of biotin-streptavidin to detect the biotin coupled at these carboxylases. We observed that the depletion of the key AID fusions resulted in a dramatic decrease in streptavidin fluorescence detected by both Western blots and IFA after 24 and 36 h induction with auxin (Fig. 3f–i). We further probed the fluorescence by IFA at regular- and over-exposed conditions and

**Fig. 3 | The micropore mediates salvage of host cytosolic materials and Golgi ceramide. a–e** The key micropore proteins are involved in GFP acquisition from host cell. **a** The parasite lines were grown in GFP-expressing host cells for observation and counting of GFP+ and GFP− parasites. Percentages of GFP+ parasites in population were plotted for three independent experiments with triplicates ($n > 100$ parasites for each replicate). **b, c** Band intensities of Western blots with AID-K13 induced for 2, 4 and 6 h were normalized to those at 0 h ($N = 4$). **d, e** In GFP acquisition assays, LHVS (10 μM) was added, while the IAA induction started for the induced groups, as illustrated (**d**), three independent experiments were performed ($n > 200$ for each replicate). Scale = 2 μm. **f, g** Western blot detection of biotin levels in parasites, showing streptavidin Li-COR IRDye 800CW (Strep-800CW) bands on blots (green) (**f**). The intensities of ACC1 (288 kDa) and PC (152 kDa) were normalized to those at 0 h (−IAA) (**g**). **h, i** IFA detection of biotin levels, showing streptavidin Alexa Flour-488 (Strep-488) (green) signal in the AID lines grown in IAA, and lines grown in the vehicle (−IAA) (**h**). ACP was the apicoplast marker. The intensities of vacuoles (8 and 16 parasites) were measured ($n = 48$), and normalized to the control (−IAA), and plotted as ratio (**i**). Scale = 5 μm. **j, k** Detection of ceramide salvage by pulsing parasites with the probe, showing representative images of the pulsing with BODIPY TR Ceramide for 30 min in parasites grown in auxin for 36 h (**j**).The staining foci numbers and fluorescence in vacuoles were measured automatically by the software system ($N \geq 50$ for each independent experiment) (**k**). Scale = 10 μm. Three independent experiments and triplicates were performed, and **a, c, e, g, i, k** Mean ± SD, and analyzed by two-way ANOVA with Tukey's multiple comparison, ****$p < 0.0001$ and other exact $p$ values with significance are provided. **b, c, f, g** The samples were derived from the same experiment and blots were processed in parallel with the loading control on the same blots.

observed clear perturbations of biotin levels in the apicoplast and mitochondrion at both exposure conditions in AID-K13 (Supplementary Fig. 6b). Biotin level dropped at 12 h and almost reached the basal level at 18–30 h as shown by Western blots at specific induction time points (Supplementary Fig. 6c). Additional experiments showed that ACC1 and PC remained at the same protein levels in the AID-K13 line grown in auxin, when compared with the vehicle, for 36 h (Supplementary Fig. 6d), further supporting the association of parasite biotin level with the micropore proteins.

To detect the salvage of ceramide, fatty acids and cholesterol, we utilized fluorescent probes to pulse parasites grown in auxin for 36 h, which enabled the subsequent tracking of probe-derived compositions, as reported previously[6,17,63]. Pulsing with a BODIPY TR ceramide-BSA complex showed that the depletion of AID fusion proteins led to fewer fluorescent foci per parasite vacuole, as well as a significantly lower mean intensity distributed in single parasites (Fig. 3j, k). In contrast, the depletion of the key AID fusions had no effect on the salvage of fatty acids and cholesterol, while pulsing with BODIPY FL C12-BSA and TopFluor cholesterol-LDL (without lipoprotein in the growth media) (Supplementary Fig. 6e–g), respectively. As the mitochondrial lipoic acid is derived from salvage from the host cell[23], we tried to detect coupled lipoic acid by Western blotting, but did not see any obvious changes in parasites when grown in auxin (Supplementary Fig. 6h). Taken together, the Kelch13 complex has a crucial role in nutrient salvage from the host cell cytosol and Golgi apparatus.

### The micropore proteins maintain fitness of extracellular parasites

To examine parasite morphology and cellular structures, we performed immunofluorescence microscopy on the inner membrane complex (IMC1, MLC1 and GAP45), the mitochondrion (Hsp60) and the apicoplast (ACP) by staining with the corresponding antibodies in the representative AID line of AID-K13 (Supplementary Fig. 7). The differential interference contrast (DIC) showed clear changes of the parasite rosette orientation in the K13-depleted parasites, which was further supported by fluorescence staining of the parasites (Supplementary Fig. 7). In contrast to no obvious changes of the IMC and the mitochondrion, the apicoplast appeared to be enlarged in K13-depleted parasites (Supplementary Fig. 7). This apicoplast phenotype is likely to be resulted from the reduced biotin level, as biotin is the key factor for the activity of ACC1, the key enzyme for fatty acid biosynthesis in the apicoplast. To test parasite viability, we assayed parasite egress and invasion. We observed egress defects at 5 and 10 min after egress stimulation by A23187, and observed a significant increase at 10 min, compared 5 min, when the key AID fusion parasites were grown in auxin for 36 h (Supplementary Fig. 8a). Interestingly, freshly egressed extracellular parasites had the full capability to invade host cells after growth in auxin for 36 h (Supplementary Fig. 8b). In contrast, the same extracellular parasites dramatically lost invasion capability following incubation in DMEM at 37 °C for 3 h (Supplementary Fig. 8c). We then wondered if fitness of extracellular parasites was being affected by the depletion of the micropore proteins. We used probes from a live/dead cell imaging kit to detect healthy and unhealthy cells by staining the parasites green and red, respectively. We observed that the red staining significantly increased in the key AID lines after induction for 36 h. This remained true for the growth-affecting AID lines, except for MPP1-AID (Supplementary Fig. 8d, e), and was consistent with the results of the GFP uptake. In summary, the key AID fusion lines grown in auxin for 36 h were still viable but lost fitness in the extracellular environment.

### The micropore is associated with parasite metabolic pathways

To gain insights into the global impact of the micropore on the parasite, we performed transcriptomics using RNA-seq and untargeted metabolomics using chromatography-mass spectrometry (LC-MS) and gas chromatography-MS (GC-MS) with AID-K13, tKD and TIR1 lines grown in auxin. To analyze overall changes, 6 of the experimental groups (the EPS15-AID, AID-K13 and tKD lines induced for 12 and 36 h) were compared with the control TIR1 using the full transcriptome (Supplementary Data 10) and differential transcriptome (Supplementary Data 11). To conduct the analysis, we used two R-based tools: Weighted Gene Correlation Network Analysis (WGCNA) and the fuzzy c-means algorithms[64,65], respectively. The WGCNA identified three modules with significantly higher or lower expression of genes in the TIR1 control, when compared with the AID induced groups, thereby revealing several metabolic pathways associated with the micropore (Fig. 4a and Supplementary Data 12). The fuzzy c-means algorithms identified 4 characteristic transcript profiles across the AID groups (Supplementary Fig. 9a), from which we observed the related pathways and biological functions of glycolysis and the TCA cycle (Supplementary Data 13 and 14). Overall, the transcriptomics revealed that there were significant changes in critical metabolic pathways upon depletion of the micropore proteins.

We then employed untargeted metabolomics and detected differences in ~50 (GCMS) and ~100 (LCMS) metabolites (Supplementary Data 15 and 16). Comparing the AID lines vs. TIR1, grown in auxin for 36 h, the metabolomics identified, respectively, 26 and 71 differential metabolites in the first batch of measurements (experiment 1, $n = 5$) and in another biological replicate batch of measurements (experiment 2, $n = 3$) (Supplementary Data 17 and 18). The differential metabolites were subjected to enrichment analysis, which revealed similar pathways in both of the experiments (Fig. 4b and Supplementary Data 19). In comparison, experiment 2 additionally enriched carbon-based metabolism of the TCA cycle and glycolysis (Fig. 4b). Additionally, experiment 2 also identified five differential metabolites from the same pathway—the TCA cycle (Fig. 4c and Supplementary Data 18). Notably, both experiments identified gamma-aminobutyric acid, providing further support for there being a likely defect in the TCA cycle in the parasites. What is noteworthy is that both experiments clearly found ceramide species as differential metabolites (Fig. 4d), supporting the previous results from the uptake of exogenous ceramide. Unexpectedly, variations between the two experiments were

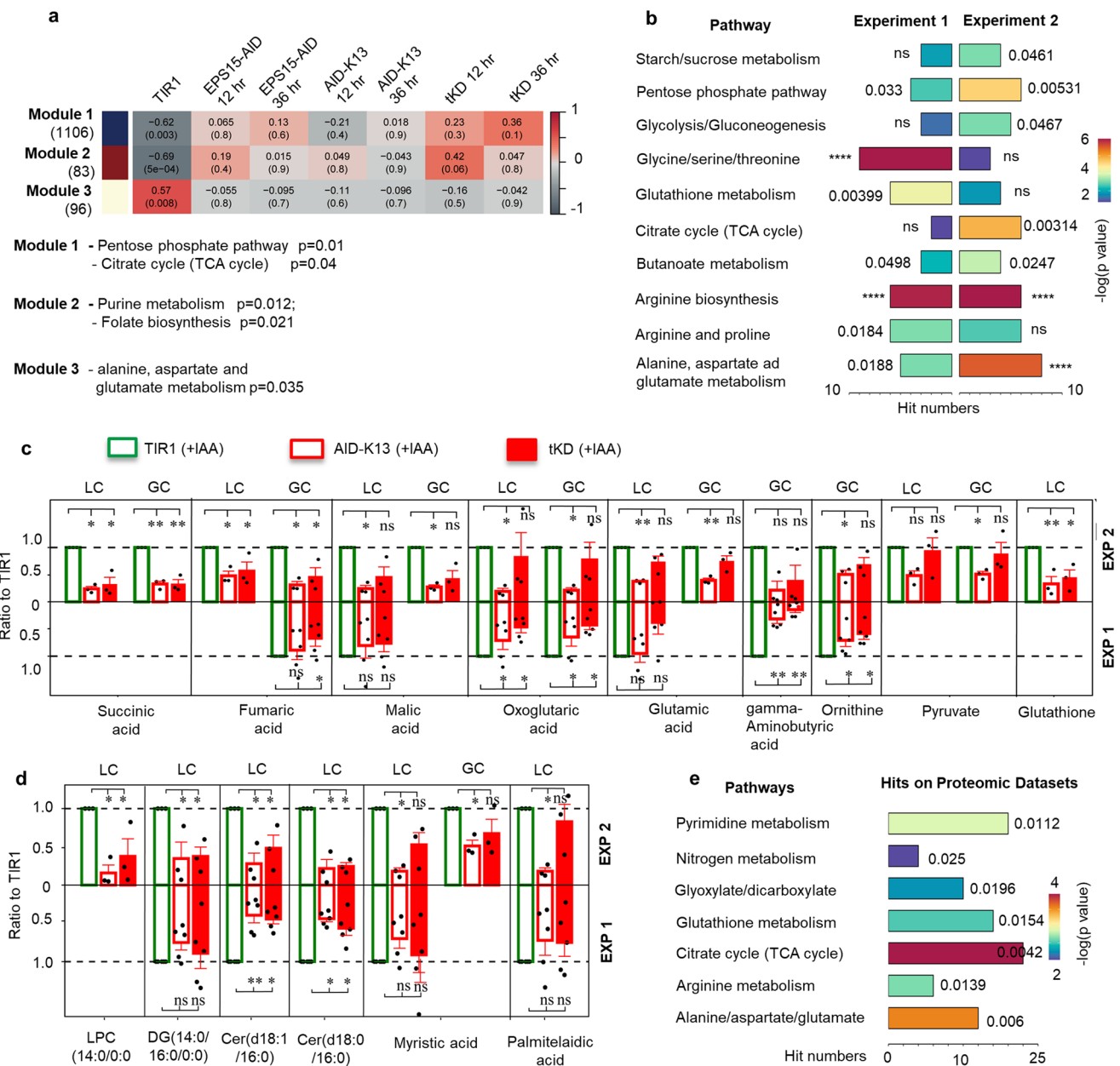

**Fig. 4 | The micropore proteins are associated with metabolic pathways.**
**a** Expression pattern analysis on *T. gondii* mRNA transcripts generated with +IAA induced lines. The WGCNA analysis identified characteristic modules with high or low expression of genes in TIR1, which reflected overall changes in the AID lines. The statistical analysis was performed based on hypergeometric test. Absolute *R* value (upper value) >0.55 and *p* < 0.05 (brackets) in TIR1 were set as the threshold of significance. The enrichment of pathways are shown for significant modules 1–3. **b** Metabolic enrichment with differential metabolites identified in experiment 1 (*n* = 5) and in experiment 2 (*n* = 3). Parasite lines (TIR1, AID-K13 and tKD) were grown in auxin for 36 h in parallel, and purified either at room temperature (Experiment 1) or in chilled PBS on ice (Experiment 2), and analyzed in two independent experiments by LCMS and GCMS. **c, d** Major metabolites identified in untargeted metabolomics of LC-MS (LC) and GC-MS (GC). The metabolites identified in the AID lines from both Experiment 1 (EXP 1) and Experiment 2 (EXP 2) were normalized to the

corresponding TIR1. The (empty or filled) color columns denote the metabolites identified in the parasite lines both in EXP1 and in EXP 2, while the empty boxes (without the color columns) in EXP 1 indicate the metabolites that were not identified by LC or GC. Cer ceramide, LPC lyso-PC, DG diacylglycerol. *p < 0.05; **p < 0.01; ns not significant. **e** Enrichment of metabolic pathways with datasets extracted from the proximity biotin labeling in Fig. 2c. Criteria for the dataset extraction: 0 peptides in the control (*N* = 4) and ≥2 peptides from either of the TurboID fusions (*N* = 2 for each) (see Supplementary Data 20). The enrichment analysis was performed by ClusterProfiler in R. **b**–**d** The data were shown with mean ± SEM (**c, d**), and were analyzed by one-way ANOVA with Dunnett's multiple comparison. **b, e** Hits represented the numbers of metabolites aligned to the corresponding pathways, while column colors indicated –log(*p* value); the enrichment was statistically analyzed by Hypergeometric test; ****p < 0.0001 and other exact *p* values with significance are provided; ns not significant.

observed, which might be caused by batch differences (two biological replicates) and sampling temperature differences (room temperature in experiment 1 vs. 4 °C in experiment 2) between experiments. In addition, variation between the AID lines were also observed, which we suspected was derived from the additional depletion of proteins in tKD (EPS15 and PPG1). This strain difference could explain the lower level of

readouts in tKD vs. AID-K13 in experiment 1. However, it has to be noted that one of the readouts (*n* = 3) in tKD in experiment 2 was generally higher than the others. Thus, we consider that the higher level of metabolites in experiment 2 is likely to have resulted from experimental difference but not from strain differences. Moreover, the drop in the lipids, such as lyso-PC and diacylglycerol (DG) (Fig. 4d),

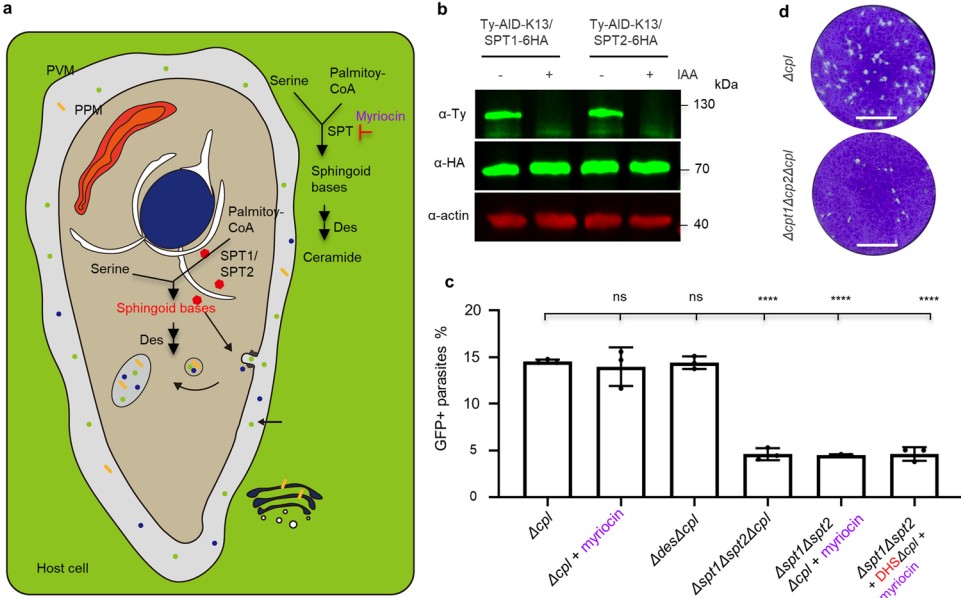

**Fig. 5 | The micropore endocytosis requires ceramide de novo synthesis.**
**a** Model of ceramide de novo synthesis in the endoplasmic reticulum (ER) and Golgi of parasites and host cells. Sphingoid bases include 3-ketodihydroxysphingonine (KDS) and dihydrosphingosine (DHS), sequentially catalyzed by SPT and a reductase, and finally forming ceramide by means of a desaturase (Des). **b** Levels of SPT1 and SPT2 showed no changes upon depletion of Kelch13. SPT1 and SPT2 were separately tagged with 6HA in the AID-Kelch13 line, and the lines were grown ±IAA for 36 h. **c** Parasite ceramide de novo synthesis pathway is required for the micropore. The lines Δcpl, Δspt1Δspt2Δcpl and ΔdesΔcpl were used for GFP acquisition tests. Myriocin (10 µM) was used to inhibit the host SPT, and external DHS (10 µM) was supplemented in the Δspt1Δspt2Δcpl. Data are shown as a mean ± SD from three independent experiments and triplicates (N ≥ 100 for each replicate), and analyzed by one-way ANOVA with Tukey's multiple comparison. ****$p < 0.0001$; ns not significant. **d** Plaque formation by the knockdown lines of Δspt1Δspt2Δcpl and Δcpl. Scale = 0.5 cm.

may be a result of the perturbation of biotin levels or blockage of vesicle membrane flow in the micropore PPM.

We combined the -omics information to analyze the defects in central carbon metabolism and found that the major changes were probably both transcriptional and biochemical (Supplementary Fig. 9b). To confirm the metabolic defects, we tested parasite replication using different carbon sources. It showed that the key AID lines treated with auxin for 30 h grew less well in glucose as the sole carbon source, than in the full medium (D5) or medium supplemented with glutamine (Supplementary Fig. 10a). We further tested the mitochondrial membrane potential using mitotracker red and found that it was undetectable in AID-K13 in both D5 and the media containing glucose when grown in auxin. In contrast, it recovered in the media containing glutamine when grown in auxin (Supplementary Fig. 10b). These results suggested that the parasites depleted with proteins could not efficiently utilize glucose, and, instead, switched to a rewiring of cellular metabolism with a reliance on glutaminolysis. This finding is consistent with the notion that glucose is the major carbon source for tachyzoites, and with the notion of cooperation with glutamine under when exposed glucose limited conditions[66,67]. To explain the defects, we re-analyzed the MS datasets of the biotinylated proteins (Supplementary Data 20) and found that the hits were enriched in the KEGG pathways in a comparable way to the enrichment seen with the differential metabolites (Fig. 4e and Supplementary Data 21). These results indicate potential protein interactions between the micropore proteins and metabolic enzymes, which, coincidently, have been observed in artemisinin resistance lines in *Plasmodium*[68,69].

**Micropore maximal activity requires the ceramide de novo synthesis pathway**

*T. gondii* can synthesize ceramide de novo, where SPT1 (TGGT1_290980) and SPT2 (TGGT1_290970), of prokaryotic origin, are the first and rate-limiting enzymes, and Des (TGGT1_237200) is the last

maturation enzyme[21,70] (Fig. 5a). To study any potential effects of depletion of Kelch13 on ceramide de novo synthesis, we tagged SPT1 and SPT2 in the AID-K13 line, and observed no changes with the protein levels of SPT1 and SPT2 by Western blotting (Fig. 5b). This result indicated that the synthesis did not contribute to the decrease of ceramide observed in the metabolomics assays. Previous studies have shown that sphingoid bases, early metabolites of this synthesis process, regulate CME in mammals and yeast[71,72]. We generated knockout lines (i.e., spt1, spt2 and des), in the background of Δcpl, and also used myriocin to inhibit the host SPT[21]. The aim of this approach was to study potential regulation of micropore endocytosis. We observed that GFP acquisition was significantly affected in Δspt1Δspt2Δcpl and dropped to one third of the control, whereas this result was not observed in ΔdesΔcpl nor following in the inhibition of host cells by myriocin (Fig. 5c). This result suggested that, for maximal micropore activity, the ceramide de novo synthesis pathway is required in the parasite but not in the host cell. The addition of an early metabolite, dihydrosphingosine (DHS), did not recover the defective GFP acquisition in Δspt1Δspt2Δcpl (Fig. 5c), suggesting that early metabolites, such as DHS, were unable to gain access to the parasites. The drop in the extent of GFP uptake in the Δspt1Δspt2Δcpl was similar to that found during endocytosis in knockout mutants of yeast[71]. We further analyzed plaque formation in the Δspt1Δspt2Δcpl line, which showed obvious growth defects, when compared to the Δcpl line (Fig. 5d). A recent study reported that the double knockout of spt1 and spt2 resulted in defects of rhoptry discharge, causing a clear in vitro growth defect[73]. As a result, it was unknown to what extent the growth defect was derived from regulation by micropore endocytosis. However, the question of how the ceramide de novo synthesis pathway impacts endocytosis remains unknown, although, in yeast, it has been proposed to act on actin via a kinase-phosphatase system[74,75]. In summary, these results support the notion that the micropore mediates the salvage of host ceramide and requires the parasite de novo synthesis pathway.

## The Kelch13 complex is associated with the integrity of the micropore

We wondered if the Kelch13 complex plays a role in the integrity of the micropore. We then examined the micropore in intracellular parasites by transmission electron microscopy (TEM) in the key AID lines (EPS15-AID, AID-K13 and tKD) and TIR1 lines grown in auxin for 30 h. Serial TEM showed that the micropore lost the invagination of the PPM and instead it produced a new invagination of the IMC or disruption of the IMC (Fig. 6a and Supplementary Fig. 11). The IMC disruption occasionally contained leakage of cytosolic contents (Supplementary Fig. 11). We then examined 600 parasites/line using conventional TEM and quantified the micropore phenotype and the defects described above. This analysis was extended to AID-PPG1 and UBP1-AID to enable comparisons with the key AID lines. The quantification assay showed that the normal micropore was unable to be found any more, instead only the defective forms were identified in the key AID lines (Fig. 6a). In contrast, deformed micropores with a shrunken base were seen in the UBP1-AID line, while morphologically normal micropores and increased numbers were observed in the PPG1-AID line (Fig. 6a). These findings supported the potential localization of UBP1 at the base of the micropore and the involvement of PPG1 in tethering it to the cytoskeleton, respectively. Intriguingly, the TEM identified an unusual PPM invagination structure close to the apex and a stuck vesicle inside the structure in the key AID lines but not in the UBP1 and PPG1 parasites. In contrast, a similar invagination was seen only in one case in the TIR1 parasite and that case had no stuck vesicle (Fig. 6b and Supplementary Fig. 12). This unusual invagination has been previously observed in parasites grown in 200 μm oleic acid[59], implying that the newly emerged invagination was likely to be induced by the accumulation of lipids in the PV. Of importance, the quantified TEMs carried out on the key AID lines showed similar numbers for both the apically emerged invagination and the micropore defects (Supplementary Fig. 12b and Fig. 6a), indicating that these were concomitant features.

We further examined the micropore in extracellular parasites, using scanning electron microscopy (SEM), and observed that the micropores were corrupted in the key AID lines when grown in auxin and, in many cases, possessed a hanging membrane (Fig. 6c). Quantification of these observations confirmed that all the micropores were corrupted, and the majority of them contained hanging membranes (Fig. 6d). Size measurements revealed that depletion of key proteins caused significant destruction of the micropore by causing a size increase ten times larger in the AID lines than the normal one in TIR1 (Fig. 6e). These results suggested that the parasites depleted with these proteins were leaky, and could explain the loss of fitness in the extracellular parasites as observed above. Collectively, these EM results proved that Kelch13 and its interactors are associated with the integrity of the micropore in *T. gondii*.

## Kelch13 is essential for parasite virulence of *Toxoplasma* in mice

We next examined the parasite virulence of the AID fusion parasites in mice administered with auxin. First, we tested whether our study had correctly dissected the proteins in vitro using complementation of the AID-Kelch13 line with HA-Kelch13. The HA-Kelch13 showed the correct localization and there was no change in auxin vs. vehicle treatment, however the endogenous AID-Kelch13 was depleted by growth in auxin (Fig. 7a, b). We observed that the plaque defects in the AID-Kelch13 line could be complemented by HA-Kelch13 (Fig. 7c). Intraperitoneal infection was carried out in mice with TIR1, the key AID lines, AP2αc-AID and the complementation line. They were subsequently administered orally and intraperitoneally with either vehicle or auxin on a daily basis as reported previously[76,77]. Both mice receiving the vehicle and those mice infected with TIR1 receiving auxin succumbed to lethal toxoplasmosis by day 10 post-infection (Fig. 7d). Conversely, the auxin treatment ultimately prevented the acute virulence caused by the AID-kelch13 and tKD lines and postponed that of the EPS15-AID and AP2αc-

AID lines. However, the virulence loss in AID-Kelch13 was rescued by HA-Kelch13. These results indicated that the outcomes of this study were robust and that Kelch13 was essential for parasite growth and virulence in mice (Fig. 7d).

## Discussion

Apicomplexan parasites have evolved diverse strategies to survive in intracellular niches, including the ability to acquire nutrients from diverse cellular compartments in host cells. However, the mechanism of nutrient scavenging remains elusive. Here, we show that the *Toxoplasma* micropore is involved, in selective nutrient salvage from complex host endo-membranous compartments. Endocytosis at the micropore is mediated by a protein network—the endocytosome, where Kelch13 acts as a central protein. Intriguingly, the micropore is associated with carbon metabolism and, for maximal endocytosis activity, it requires de novo synthesis of ceramide by the parasite. Depletion of Kelch13 resulted in another invagination of the PPM, which is potentially involved in nutrient salvage, however depletion of Kelch13 removed parasite virulence, indicating that the micropore is critical for growth/virulence of *T. gondii* both in vitro and in vivo.

The micropore is ultra-structurally conserved in almost all apicomplexans and related parasites examined by electron microscopy[25,29,36,78]. It was initially hypothesized to be a feeding pore[36,78], yet few studies have addressed its function in parasites that reside in nucleated cells, although more is known about its role in the erythrocytic stages of parasite *Plasmodium*[37,79]. In addition, the absence of caveolar endocytosis[80], and the presence of the canonical endocytosis proteins, which are associated with organelle biogenesis, have caused the existence of endocytosis in *T. gondii* to be questioned[41–45]. Therefore, the classical proposal is that *T. gondii* solely relies on the activity of transporters at the PPM to acquire nutrients after digestion of internalized cellular structures in the PV[5]. Recent studies have identified the existence of an endo-lysosomal compartment and uptake of host cytosolic proteins by *T. gondii*, implying that there likely exists an endocytic process in the parasite[16,81,82]. Though progress on the cytostome in *P. falciparum* has been made recently, the composition and functions of the micropore in parasites residing in nucleated cells remains unknown. Therefore, this study has provided direct and robust data defining the micropore as the entry point of the endocytic processes in the apicomplexan model organism *T. gondii*.

In the erythrocytic forms of malaria parasites, the cytostome is considered to be the specialized structure of the micropore, which appears to be a flask-like structure containing invagination of both the PVM and PPM[33,36,83]. This cytostome is capable of efficient engulfment of semiliquid and highly homogenous hemoglobin, a conspicuous biological process[33,36,79]. This process has attracted intensive attention[79]. A recent study has identified the protein composition and, importantly, discovered its association with artemisinin resistance[49]. Compared to the *Toxoplasma* micropore, the cytostome contains additional proteins (e.g., KIC1/3/5/6/7/8/9) and more of these are essential (5 out of 14) (Supplementary Table 1), which is likely due to its specialization for hemoglobin uptake. Intriguingly, our proteomic data showed that clathrin was not present in the protein interactome, echoing the proteomic data on the *P. falciparum* cytostome[41,49]. However, another striking difference between the *Toxoplasma* micropore and *Plasmodium* cytostome lies with Kelch13, which is important only in the ring stages of *Plasmodium*, but not in the trophozoite stage that is the most active stage for hemoglobin uptake[49,79]. In contrast, our study has clearly pinpointed Kelch13 and its interactors at the micropore using immuno-EM, TEM and SEM. Our data support the fact that Kelch13 functions as a protein hub for function of the micropore that is essential to the parasite, and associates with the function of the micropore in *T. gondii*.

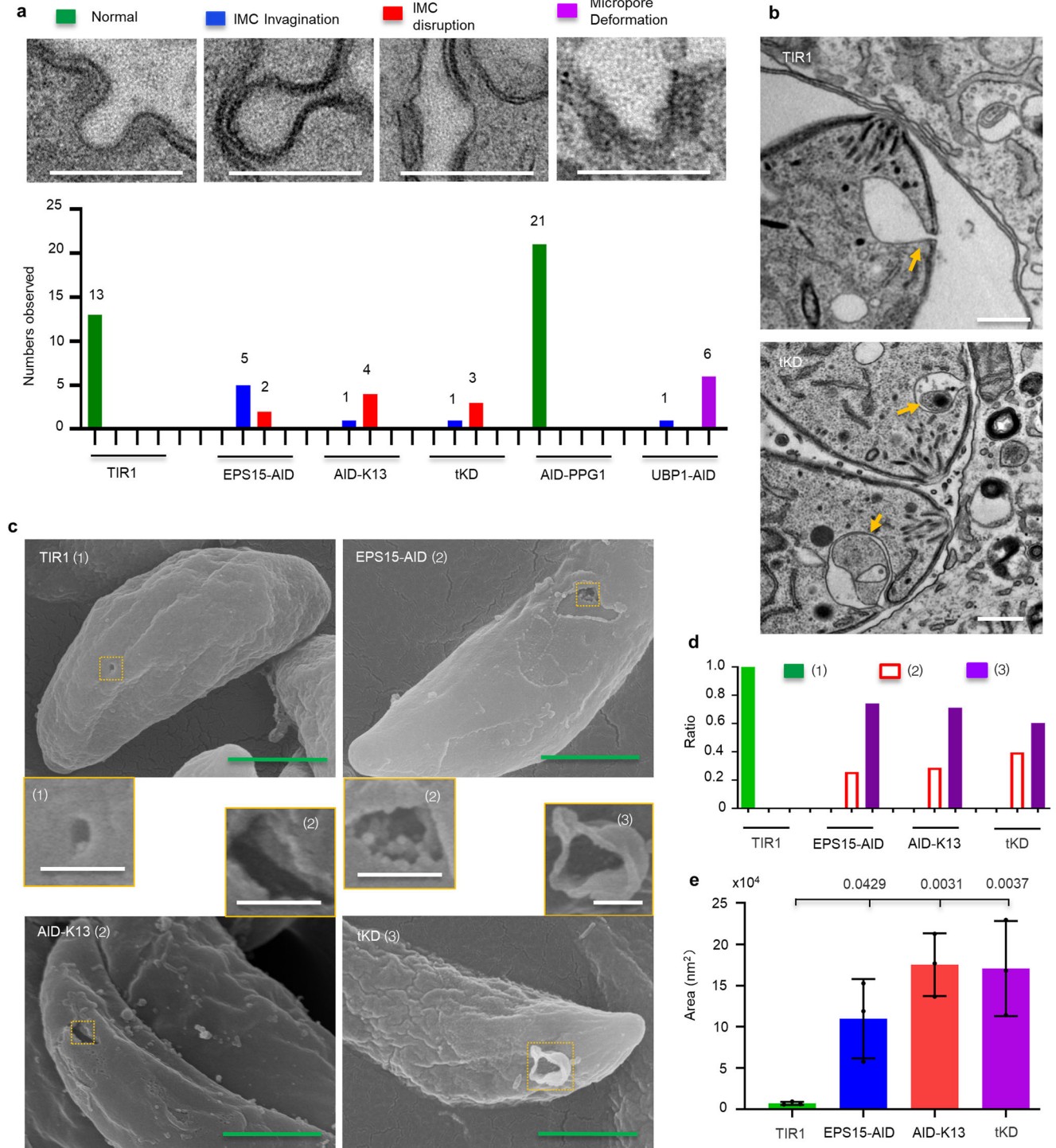

**Fig. 6 | The Kelch13 complex affects the integrity of the micropore. a** Micropore structures on TEM and its quantifications. Serial TEM was performed to examine different types of micropore defects in the parasite lines grown in auxin for 30 h. These defects were quantified by TEM for 600 parasites from each line. The micropore defects were indicated by colors, which relate to the graphs. Scale = 200 nm. **b** Plasma membrane invagination close to the apex was observed on TEM analysis in IAA induced parasites for 30 h, as in **a**. This invagination (yellow arrowhead) was previously shown in parasites treated with 200 μm oleic acid[60]. Here it shows the tKD and TIR1 grown in auxin. Scale = 500 nm. **c**, **d** Scanning microscopy observation of the corrupted micropore in parasites with IAA for 30 h.

The different types of micropore were shown with number 1, 2 and 3 in brackets, the type 1 was normal, while others were corrupted. The type 3 contains hanging membranes (**c**). These micropores were scored and calculated as ratios in the parasites where the micropore or its defects were observed (**d**) ($N \geq 40$ for each line). Green scale = 1 μm, white scale = 200 nm. **e** Area analysis of the SEM defects as shown in **c**. Data were shown with mean ± SD, and analyzed by one-way ANOVA with Tukey's multiple comparison, the exact $p$ values are provided. Three independent experiments were performed with measurements of $n > 12$ (defective) micropores in each independent experiment.

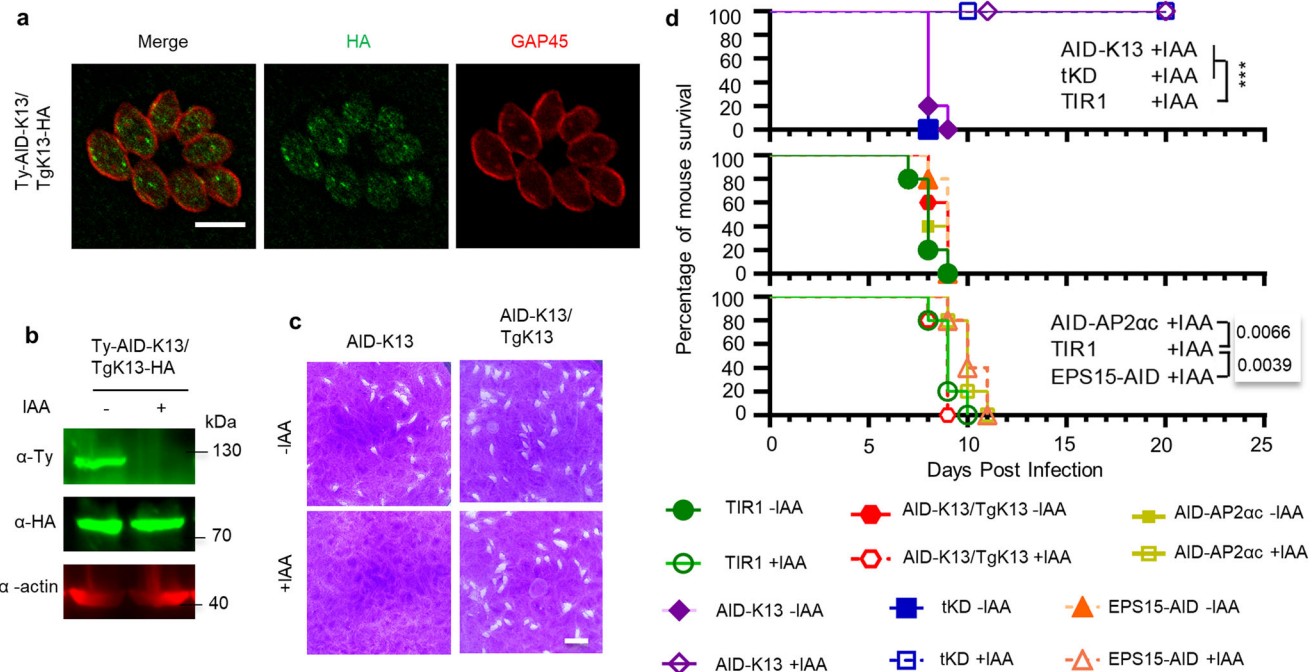

**Fig. 7 | Kelch13 is essential for parasite virulence in mice. a, b** IFA and western blot detection of complemented HA-TgKelch13 in the AID-Kelch13 line by growth in ±IAA for 24 h. Actin served as a control. Scale bar = 5 μm. **c** Plaque formation by the AID-K13 and complementation lines grown in ±IAA for 7 days. Scale bar = 0.5 cm. **d** Survival of mice infected with TIR1, AP2αc-AID, EPS15-AID, AID-K13, tKD and the complementation line (AID-K13/TgK13) in the absence (with vehicle) and presence of auxin. Mice survival was monitored for 20 days post-infection. One representative result was shown from three independent experiments, and 100 parasites were used per mouse for infection. Survival data were analyzed by the Gehan-Breslow-Wilcoxon test, comparing to the TIR1 +IAA. ***$p < 0.0001$, and $p$ values are provided; not significant for AID-K13/TgK13 +IAA vs. TIR +IAA.

Here we demonstrated that the *Toxoplasma* micropore is able to bud a single membraned endocytic vesicle for salvage of host cytosolic materials and Golgi ceramide. These nutrients are internalized to the PV by the ESCRT mechanism (with host cytosolic materials) or host Rab proteins (with Golgi ceramide) prior to the micropore endocytosis[4,6]. As fatty acids and cholesterol were internalized to the PV by H.O.S.T from other host organelles[5,59], but not scavenged by the parasites via the micropore, the micropore appears to be selective for nutrients from the PV lumen. It is noteworthy that the micropore buds vesicles, producing membrane flow, which potentially affects membrane homeostasis. Meanwhile, our proteomic analysis has not identified the potential receptors that bind to the cargos required for initiation of endocytosis. As a result, it is uncertain whether micropore endocytosis is a receptor-mediated process or a fluid uptake system. Further work is needed to clarify the initiation mechanism of micropore endocytosis.

Our study demonstrated that the micropore is associated with parasite carbon metabolism. Our metabolomics data, especially with experiment 2, pointed to the association of the TCA cycle and related pathways with the micropore proteins. This notion was confirmed by further experiments with parasites cultured in different carbon sources. Intriguingly, comparable metabolic defects were also observed in a recent study in *Plasmodium* artemisinin resistance lines that contain mutations in Kelch13[69]. These similar metabolic defects contributed to artemisinin resistance in *T. gondii* as well as in *P. falciparum*, together with mitochondrial heme[84,85]. In addition, Kelch13[C672Y] was able to confer artemisinin resistance in *T. gondii*[84], and this is expected to be associated with reduced mitochondrial activity. However, the resistance in *P. falciparum* was proposed to be mainly associated with reduced protein levels caused by mutations in Kelch13 and subsequent endocytic activity[49,79]. It is noteworthy that our study did not detect amino acids present in the differential metabolites except for glutamate. Glutamate can be synthesized de novo from metabolites of the

TCA cycle[66,86], and this probably explains its presence within the differential metabolites. However, it is very different to the scenario found in *P. falciparum*, where hemoglobin is the major source of amino acids[37,49,87]. It has been demonstrated that *T. gondii* acquires essential amino acids by transporters at the PPM[13,86,88]. Therefore, it would be highly plausible that nutrients acquired via the micropore may contribute to parasite growth to a limited extent, due to flexibility in nutrient salvage vs. synthesis in the parasite[18,22]. Nevertheless, studies suggest a conserved impact on mitochondrial metabolism with both the *Toxoplasma* micropore and the *Plasmodium* cytostome. Furthermore, we are uncertain whether the defects of mitochondrial metabolism in *T. gondii* are the result of protein interactions or the downstream effect of nutrient acquisition reduction, as reported in *P. falciparum*[68,69,89]. In this sense, our AID-K13 and complementation lines will potentially have future prospects for studies on artemisinin resistance from the angle of a homologous structure in *T. gondii*. In summary, these results suggest that the micropore affects the metabolic activities that regulate the parasite growth.

Intriguingly, we observed that ceramide de novo synthesis acts on the micropore activity to regulate endocytosis. Biochemical studies have shown that the *T. gondii* enzymes SPT1 and SPT2 catalyze the synthesis of sphingoid bases[70,90]. However, it is unknown to what extent the SPT enzymes contribute to parasite growth via the regulation of the micropore endocytosis, as the enzymes act on secretion of rhoptry proteins[73]. Nevertheless, sphingoid bases act on the CME via the actin filaments in mammals and yeast[71,72]. In *Plasmodium* parasites, actin plays a key role in hemoglobin uptake[35,91]. In addition, in *Plasmodium*, SPT (PF3D7_1415700) has a very low fitness score (PlasmoDB), although the de novo synthesis seems not to be essential[92] and some of the genes, such as Des, are not identifiable in the genome. These analyses thus imply that PfSPT may be involved in hemoglobin uptake and artemisinin resistance—a major concern for combating global malaria[79].

Despite the divergence of apicomplexan species, the domains of Kelch13 are highly conserved in the phylum, indicating that Kelch13-binding capability and functions may be highly conserved among the apicomplexans. This study provides direct and robust evidence supporting the notion that the micropore performs an essential function in controlling the endocytosis of nutrients from host complex cellular compartments, in contributing to metabolism and parasite growth and in maintaining fitness of extracellular parasites. Given the conserved nature of the micropore, the unique endocytosis mechanism, controlled by Kelch13, is likely to control similar essential processes at the micropore in other apicomplexan parasites and evolutionarily related parasites.

## Methods

### Parasite and host cell culture

The previously described lines RH$\Delta$ku80$\Delta$hxgprt[93] and RH$\Delta$ku80$\Delta$hxgprt/TIR1[58], referred to as RH$^{ku80}$and TIR1, were used to generate the transgenic lines (T. gondii tachyzoites) reported here. The parental lines and derived lines (Supplementary Data 1) were grown in human foreskin fibroblasts HFF-1 (ATCC SCRC-1041) in D5 media composed of Dulbecco's modified Eagle medium (DMEM) (12800-082, Thermo Fisher Scientific), supplemented with 5% heat-inactivated fetal bovine serum (10099-141, Gibco), 2 mM glutamine (G0200, Solarbio Biotech) and 100 units penicillin-streptomycin (P1410, Solarbio Biotech) at 37 with 5% $CO_2$. HFF-1 cells were cultured in the same media and conditions, and were all maintained as mycoplasma negative[94]. The TIR1 parental line, and their derived AID lines were cultured in HFF monolayers with 500 mM auxin (I2886, Sigma-Aldrich) (+IAA) or 0.1% ethanol alone (−IAA) for phenotypic assays, as previously described[46,58]. Parasites were allowed to naturally egress or were mechanically lysed by passing through 22 g needles, and harvested by filtration through 3.0 micron polycarbonate membranes, from which extracellular parasites were used for experimental assays.

### Animals

Six-week old female BALB/C mice were used for in vivo T. gondii infection studies, and housed under specific pathogen-free (SPF) conditions in filter-top cages and sterile water and food were provided. Mice were monitored on a daily basis to record their health by checking their appearance, bodyweight and responsiveness, and sacrificed by the endpoint. The mouse experiments were conducted according to the guidelines and regulations issued by the Veterinary Office of the China Agricultural University (Issue No. AW11402202-2-1). Female New Zealand rabbits (2–3 month old) housed in specific pathogen-free (SPF) conditions were used for antibody generation and the guidelines and regulations (Issue No. AW11402202-2-1) were applied to the rabbit experiments.

### Plasmid design and construction

All plasmids (Supplementary Data 2) were generated by a Basic Seamless Cloning and Assembly kit (CU201-02, TransGen Biotech) using existing plasmids as templates. Briefly, the pCas9-sgRNA plasmids were generated using a previously described plasmid (Addgene #54467) as the template, and three fragments were amplified using three pairs of primers named as Cas9-F1/R1, Cas9-sgRNA xx/Cas9-R2 and Cas9-F2/R3 (Supplementary Data 3). The Cas9-sgRNA xx primer contains the sgRNA sequences (20 mer) which target a specific locus either upstream of the start codon (pCas9-sgRNA 5′) or downstream of the stop codon (pCas9-sgRNA 3′). The sgRNA design was selected from the prediction webpage[95]. The generic plasmids used for the amplicons targeting at the C-terminus of a gene were generated previously[46], such as pLinker-AID-3xHA-DHFR (addgene #86669), pLinker-AID-Ty-HXGPRT (addgene #86667), and pLinker-AID-3xHA-HXGPRT (addgene #86553), or generated in this study by cloning the fragment of AID-Ty or TurboID-Ty (amplified from addgene #1116904) into the backbone

of pLinker-AID-3xHA-DHFR, producing pLinker-AID-Ty-DHFR and pLinker-TurboID-Ty-DHFR. The generic plasmids used for the amplicons targeting at the N-terminus of a gene were generated using the backbone of a plasmid named pN-Ty-DHFR or pN-Ty-HXGPRT[40], by replacement of the Ty region with Ty-TurboID, Ty-AID, HA-AID, C-myc-AID, 6HA or 6Ty. Primers used for the plasmid generation were listed for each plasmid in Supplementary Data 3 and plasmids generated in this study were listed in Supplementary Data 2. The DNA regions, including epitopes, AID and TurboID in the newly prepared plasmids were checked by DNA sequencing. The plasmid for complementation of the Ty-AID- Kelch13 line was generated by cloning the Kelch13 fragment into pN-Ty-HXGPRT to replace the Ty region and was generated using primers listed in Supplementary Data 3.

### Generation of T. gondii lines

CRISPR-Cas9 tagging technology was used for generation of all lines (Supplementary Data 1) that contain endogenous fusions at the C-terminus of proteins, and at the N-terminus of proteins in the parental lines of the RHku80$^{KO}$ or TIR1 line[46]. In brief, the CRISPR/Cas9 sgRNA 3′ plasmids (Supplementary Data 2) and CRISPR/Cas9 sgRNA 5′ plasmids (Supplementary Data 2) can efficiently produce Cas9 and sgRNA to create DNA double strand breaks (DSB) in parasites, and facilitates the integration of a tagging amplicon. The amplicon for the C-terminal tagging was generated from a generic plasmid (with the name pLinker-) containing a tag (6HA, AID-3xHA, AID-3Ty, 6Ty or TurboID-3Ty, etc.) as described above, using a pair of primers L and T (Supplementary Data 3). The amplicon for the N-terminal tagging was generated from a generic tagging plasmid (with the name pN-) containing a tag (Ty-TurboID, Ty-AID, HA-AID, myc-AID) using a pair of primers M and NL. The amplicon was then combined with the corresponding CRISPR/Cas9 sgRNA 3′ or 5′ plasmid (Supplementary Data 2) and transfected into recipient lines. The drug selection was followed on the second day based on the resistance marker used in the amplicons. The resistance markers with LoxP sites were excised by transfection with pmini-Cre[96]. The cpl gene in the TIR1- AID lines was deleted using a similar CRISPR-Cas9 strategy[94]. The complementation lines were generated by direct transfection with the plasmids containing the wild type gene (Supplementary Data 2). Transfection and drug selection were performed with 25 µg/ml mycophenolic acid (M5255, Sigma-Aldrich) and 25 µg/ml 6-xanthine (X4002, Sigma-Aldrich), 200 µg/ml 6-thioxanthine (S96242, Shanghai Yuanye Biotech), or 3 µM pyrimethamine (46706, Sigma-Aldrich)[40,46]. The lines were confirmed by IFA and diagnostic PCR, as illustrated in Supplementary Fig. 3a, b, and the PCR was designed for testing the integration site and the endogenous region with primers listed in Supplementary Data 3.

### Indirect immunofluorescence assay (IFA) staining

Primary antibodies, such as the antibodies against epitopes HA (rabbit anti-HA antibodies, 71-5500, Thermo Fisher Scientific, dilution: 1:500; mouse anti-HA, anti-HA.11, clone 16B12, 901501, Biolegend, dilution: 1:500) and C-myc (mouse anti-c-myc, mAB 9E10, MA1-980, Thermo Fisher Scientific, dilution: 1:500), were commercially purchased, as described in a previous study[46]. Rabbit anti-GRA7 antibody (dilution: 1:500) was generated previously[97] and anti-Hsp60 (1:500), actin (1:500), ACP (1:500), GAP45 (1:500) and IMC1 (1:500) were newly prepared in mice or rabbits, and confirmed by IFA and western blots, as described below. Rabbit anti-Ty (BB2) was a generous gift from Prof. Phillipe Bastin, and used at dilution of 1:500. Parasites grown in HFF monolayers on coverslips were fixed by 4% paraformaldehyde in PBS, and permeabilized using PBS containing 2.5% BSA and 0.25% Triton X-100 (unless otherwise indicated). Parasites were then incubated with different combinations of primary antibodies, followed with appropriate secondary antibodies conjugated to Alexa Fluors (A-11029, A-11031, A-11036, A-11034, all purchased from Thermo Fisher Scientific, dilution: 1:1000). For extracellular parasites, freshly lysed parasites (fresh parasites) were

washed, and resuspended in PBS, then added to coverslips coated with poly-lysine, and incubated for 10 min. Adhered parasites on coverslips were washed with PBS and fixed with 4% paraformaldehyde for 10 min. Parasites were then permeabilized with 0.25% Triton X-100 in PBS and blocked using 2.5% BSA. Antibody staining followed the protocol described above when necessary. ProLong Gold Antifade Mountant with or without DAPI (36930 and 36931, Thermo Fisher Scientific) were used to mount the coverslips on glass slides. Parasites were then imaged using a Nikon Ni-E microscope C2+ equipped with a DS-Ri2 Microscope Camera. The analysis of protein co-localization was performed using the Ni-E software system NIS Elements AR.

IFA detection of biotin level was performed with streptavidin Alexa Fluor-488 (LICOR, 926-3230), and vacuoles containing 8 and16 parasites were used for fluorescence analysis. The fluorescent intensities in vacuoles were automatically measured using the NIKON NIS Elements AR software system. At least 16 vacuoles were analyzed by the system for each independent experiment, and three independent experiments were performed.

### Stochastic optical reconstruction microscopy (STORM) imaging

Parasites were grown in HFF monolayers on glass bottom dishes (MatTek, Cat#P35G-1.5-14-C) for 24 h and fixed with 4% paraformaldehyde (PFA) (P1110, Solarbio) and 0.1% glutaraldehyde (18428-5, PolySciences) in PBS (pH 7.4) for 10 min. This was followed by washing off the excess PFA and reducing unreacted aldehyde groups with 0.1% sodium borohydride (NaBH4) in PBS. After blocking and permeabilizing in a PBS buffer containing 3% BSA (A2000, IgG free, Protease-Free, Jackson ImmunoResearch) and 0.2% Triton X-100 for 30 min, parasites were incubated with mouse anti-Ty (BB2)[98] and rabbit anti-TgIMC1 antibodies. The samples were incubated with goat anti-mouse IgG (H + L) Alexa Flour-647 antibody (A21235, Thermo Fisher Scientific), and goat anti-rabbit IgG F(ab')2 fragment - atto 488 (36098, Sigma-Aldrich). After washing off the residue antibodies, parasites on the glass bottom were used for imaging using a Nikon combined confocal A1/SIM/STORM system with four activation/imaging lasers and a CFI Apo SR TIRF 100X oil (NA 1.49) objective equipped with an Andor EMCCD camera iXON 897. The STORM imaging was performed in an imaging buffer containing 50 mM Tris (pH 8.0) and 10 mM NaCl, an oxygen scavenging system consisting of 1.2 mg/ml glucose oxidase (Sigma-Aldrich), 73 μm/ml catalase (Sigma-Aldrich) with 10% (w/v) glucose, and 304 mM β-mercaptoethylamine (Sigma-Aldrich). Data were processed using the NIS-Elements AR (Nikon) software.

### Analysis of TurboID lines and purification of biotinylated proteins

The TurboID fusion lines were generated using a CRISPR-Cas9 approach as described above and grown in HFF monolayers for 30 h before treatment with 500 μM D-biotin (B4639, Sigma-Aldrich) for 1 h. Parasites were fixed, followed by the IFA steps. The parasites were first stained with mouse anti-Ty, followed by secondary anti-mouse antibodies conjugated with Alexa Fluor-568 (A11031, Thermo Fisher Scientific, dilution: 1:500), and streptavidin Alexa Fluor-488 (926-3230, LICOR, dilution: 1:500). Western blot detection was performed for the biotinylated protein samples using anti-Ty (BB2)[98] antibodies and streptavidin LICOR CW800 (926-32230, LICOR, dilution: 1:500). The purification of biotinylated proteins was carried out as previously described[99]. In brief, parasites were fully lysed in a buffer containing 1% Triton X-100, 0.2% SDS and 0.5% deoxycholate and sonicated using a microtip in 550 sonic dismembrator (Thermo Fisher Scientific). This was followed by purification of biotinylated proteins using streptavidin beads (Pierce, 88816).

### Mass-spectrometry and interactome analysis

Purified biotinylated proteins were eluted in a SDS sample buffer containing 2 μM biotin at 90 °C for 10 min and run on SDS-PAGE gels.

The protein gels were fixed and stained in Coomassie blue R250, 45% methanol, 10% glacial acetic acid, and destained in 25% methanol and 8% glacial acetic acid. The protein lanes were sliced, frozen and dried by vacuum. The dried gels were rehydrated and chopped into small pieces, followed by reduction, alkylating and washing steps to remove the stain and SDS. The samples were digested by trypsin, dried and re-dissolved in 2.5% acetonitrile and 0.1% formic acid. Each digest was run by nanoLC-MS/MS using a 2 h gradient on a 0.075 mm × 250 mm C18 column feeding into Q-Exactive HF mass spectrometer[99]. The control line RHku80[ko] and TurboID fusion lines were analyzed in parallel with two technical and biological replicates. Resulting spectra were searched against a combined database of *T. gondii* (http://ToxoDB.org, release 53), human proteins and a decoy database, using Mascot and Scaffold. The current view in Scaffold were exported into an excel spreadsheet with a setting of peptides ≥ 2, protein threshold 99% and peptide threshold 95%. The basic data file (Supplementary Data 8) was minimally edited for interactome analysis with an R-based SFINX local package. Another file containing the baits (the TurboID fusions) was created for the SFINX analysis together with the Basic data file. Resulting data, containing both *p* values and SFINX scores, were retrieved (Supplementary Data 9), and these results were used for assessing the strictness of protein-protein interactions, and identification of core interactors using parameters of a strictness value of 4 and *p* value <0.00005.

### Bioinformatic analysis

Proteins in the TOXODB were identified by searching with the words EH domain, BAR domain, and orthologs of AP2 components in *T. gondii* were identified using their mammalian equivalents. Apicomplexan orthologs were identified by searching against proteomes in VEuPathDB (https://VEuPathDB.org/veupathdb/app), and TOXODB (https://ToxoDB.org/toxo/app/) with *T. gondii* proteins as queries. Hits with an *E*-value <1e−7 were considered as orthologues (Supplementary Data 6). A heatmap of protein conservation was generated using the *E*-values in the Pheatmap package in R. Domains of proteins were analyzed with InterPro (https://www.ebi.ac.uk/interpro/). Gene ontology (GO) Terms and KEGG pathways were analyzed, in TOXODB, and items with *p* value <0.05 were considered significant.

### Antibody preparation

*T. gondii* cDNA fragments encoding full or partial polypeptides of known proteins, i.e., Hsp60, GAP45, actin, ACP and IMC1, were cloned into pET28a (Thermo Fisher Scientific) using primers to create corresponding expression plasmids (Supplementary Data 2 and 3), and transformed into *E. coli* BL21DE3 (CD601-02, Transgene Biotech). The recombinant proteins were purified using HisPur Ni-NTA spin columns (88225, Thermo Fisher Scientific), and prepared for antiserum generation in mice (5 mice per antibody generation) or rabbits (2 rabbits per antibody generation) using Inject Freund's complete adjuvant (F5881, Sigma-Aldrich), as recommended by the manufacturer' protocols. The anti-sera were tested using serial dilutions on IFA and western blots, followed by an assessment of protein localization and molecular weights of the corresponding *T. gondii* proteins.

### Western blotting

Freshly or mechanically egressed parasites were harvested by filtration through 3.0 micron polycarbonate membranes, and parasite pellets were resuspended in PBS and mixed with 5x Laemmli sample buffer. Protein samples were cooked at 95 °C, separated by SDS-PAGE and blotted using a Bio-Rad Wet-Blotting System. This was followed by incubations with appropriate primary antibodies against actin (generated in this study) (dilution: 1:500), anti-HA (rabbit anti-HA antibodies, 71-5500, Thermo Fisher Scientific, dilution: 1:500; mouse anti-HA, anti-HA.11, clone 16B12, 901501, Biolegend, dilution: 1:500), anti-Ty (BB2), anti-C-myc (mouse anti-c-myc, mAB 9E10, Life Technologies)

(dilution: 1:500), or anti-lipoic acid (rabbit antibody purchased from Abcam, ab58724) (dilution:1:250). The primary antibodies were then detected by secondary antibodies conjugated with LI-COR 800CW or 680CW reagents (926-32210, 926-32211, 926-68071 and 926-68070, all purchased from LICOR, dilution: 1:1000). Biotin levels on western blots were detected by streptavidin LI-COR 800CW (926-32230, LICOR, dilution: 1:500). Blots were then visualized using a Bio-Rad ChemiDOC MP imaging system. The intensities of western blot bands were scored by ImageJ and the background intensity was deducted for calculation of final intensities of target bands. The resulting intensities in treated samples were compared to those of control samples in the same independent experiment, generating ratios for plotting in Graphpad v8.0 (the control was considered as 1).

## Plaque formation

Freshly lysed parasites were counted using hemacytometers (Hausser Scientific), and 150 parasites were inoculated on confluent HFF monolayers in 6-well plates with addition of either 500 µM auxin or ethanol alone (0.1%) and grown in D5 media at 37 °C for 7 days. Monolayers on 6-well plates were then fixed with 70% ethanol for 15 min, stained with 0.5% crystal violet for 5 min, then gently washed with water, and dried at room temperature. Plaques formed on the wells were counted by eye and tiny plaques were counted under ×2 or ×4 magnification using a Nikon inverted fluorescence microscope TS2R-FL. The plaque monolayers were recorded by scanning using a HP Scanjet G4050, and plaque areas were calculated with the images using ImageJ 1.52p.

## Parasite replication

Parasite replication was analyzed by IFA using GAP45 antibodies, as previously described[46]. In brief, parasites were grown in 500 µM auxin, or ethanol alone (0.1%) for 24 h on monolayers in D5 media in 24-well plates with coverslips, unless otherwise stated. For testing parasite replication in media containing different carbon sources, the media were prepared with a commercial DMEM without glucose, glutamine and pyruvate (A1443001, Gibco). This was supplemented with 5% FBS and 100 unit penicillin-streptomycin, and one carbon source from either 25 mM glucose, 2 mM glutamine or 1.0 mM pyruvate. The parasites and HFF monolayers were fixed with 4% paraformaldehyde for 10 min, permeabilized with PBS containing 2.5% BSA and 0.25% Triton X-100, followed by IFA using GAP45 antibodies and secondary anti-rabbit antibodies conjugated with Alexa Fluor 568. The coverslips were then mounted using ProLong Gold Antifade Mountant without DAPI. The vacuoles containing different numbers of parasites ($N > 100$ vacuoles in each replicate) were counted under a Nikon Ni-E microscope C2+, and plotted as ratios of different types of vacuoles in the total vacuoles counted.

## Parasite egress and invasion assays

Parasite egress were performed as described in a previous study[46]. In brief, parasites were grown in HFF monolayers on coverslips in the presence of auxin or the vehicle for 36 h. The egress assay was performed with stimulation using 3 µM A23187 (%w/v DMSO) for 5 min and 10 min in DMEM at 37 °C. This was followed by IFA using primary antibodies against GRA7 (Rabbit) and IMC1 (Mouse) and secondary antibodies conjugated with Alexa Fluor-568 and 488 (reagents are used as described above). The egressed and non-egressed vacuoles were counted (>100 vacuoles in each replicate) under a NIKON microscope Ni-E C2+, by recognition of vacuolar staining with GRA7 and parasites staining with IMC1. The invasion assay was performed with freshly and mechanically egressed parasites, and these parasites incubated in DMEM at 37 °C for 3 h. Confluent HFF monolayers were challenged by the parasites, followed by centrifugation at $8 \times g$ for 1 min to bring down the parasites. The plates were immediately placed in a water bath at 37 °C for 30 min, and subsequently fixed by cold 4%

paraformaldehyde for 10 min. The monolayers were processed for IFA with mouse mAb DG52 to SAG1 without permeabilization, followed by secondary antibodies conjugated to Alexa Fluor-488. The monolayers were then permeabilized, stained with rabbit anti-GAP45 antibodies and secondary antibodies. At least 100 parasites on each coverslip with triplicates per experiment were scored.

## Generation of GFP-expressing HFF cells

The GFP expression in the HFF-1 cell line was performed using a Lentiviral expression system II. In brief, the GFP fragment was cloned into the lentivector pLVX-EF1a-IRES-puromycin (#134858, addgene) between the BamHI and EcoRI sites, creating pLVX-EF1a-GFP-IRES-puromycin (see primers and plasmids in Supplementary Data 2 and 3). This plasmid was combined with packaging plasmids psPAX2 (addgene #12260) and pMD2.G (addgene #12259) in Opti-MEM I (31985062, Gibco), and mixed with Fugene HD (E2311, Promega), and transfected into 293T cells, according to the manufacture's recommendations. After 48 h, viruses were collected in the supernatant, followed by mixture with polybrene (H8761, Solarbio)(final conc. 6 µg/ml) in a 12-well plate well containing 30–50% confluent HFF-1 cell monolayers. At 24 h of post-transduction, the HFF-1 cell monolayers were trypsinized, re-seeded on 12-well plates, and puromycin (P8230, Solarbio) (2.5 µg/ml) was added and maintained for 3 days. The GFP-expressing HFF-1 cells were examined for GFP expression using a Nikon Inverted Fluorescent Microscope TS2R-FL, and the cell monolayers were maintained and expanded in D5 media, cryopreserved, and subsequently used for GFP acquisition experiments.

## GFP uptake by *T. gondii* parasites

The GFP uptake assay was performed as described in a previous study with minor modifications[16]. In brief, TIR1/Δ*cpl* and AID lines with Δ*cpl* were grown in 500 µM auxin, or ethanol alone (0.1%) on the GFP-expressing HFF cell monolayers for 36 h. To test the direct impact of GFP acquisition in the AID-K13 parasites, parasites (AID-K13) were grown in the GFP-expressing HFF cell monolayers for 30 h in total, but induced by auxin for 0, 2, 4, or 6 h. The non-induced and induced groups were cultured in parallel and added with LHVS (SML2857, Sigma-Aldrich) (10 µM) at the corresponding induction time points, as illustrated in Fig. 3d. The parasites were mechanically egressed and harvested by filtration through 3.0 micron polycarbonate membranes. The extracellular parasites were adhered to poly-lysine-coated coverslips and fixed by 4% paraformaldehyde. The parasites were permeabilized, mounted by ProLong Gold Antifade Mountant with DAPI and subsequently visualized using a Nikon Ni-E microscope C2+ under the same exposure parameters. All extracellular parasites on images with or without GFP fluorescence (GFP foci) were counted. Note that no parasites containing diffused GFP fluorescence were observed and parasites containing GFP foci were denoted as GFP+ parasites. At least 100 parasites were counted for each replicate, and two or three independent experiments were performed with triplicates. The percentage of GFP+ in the total parasites scored was plotted in Graphpad v8.0.

## Acquisition of fluorescent lipid reagents by intracellular parasites

Acquisition of ceramide and fatty acid was performed as previously described[6,17]. In brief, parasites were grown in 500 µM auxin, or ethanol alone (0.1%) for 36 h on HFF monolayers on coverslips. The monolayers with parasites were then washed with PBS, and incubated in DMEM supplemented with the mixture of 2.5% fatty-acid free BSA (A2000, PAA Laboratory) and 1 µg/ml BODIPY TR Ceramide (D7540, Thermo Fisher Scientific) or with the mixture of 2.5% fatty-acid free BSA and 1 µg/ml BODIPY-FL $C_{12}$ (D3822, Thermo Fisher Scientific) in a $CO_2$ incubator at 37 °C for 30 min. The parasites in monolayers were washed twice with PBS, fixed with 4% paraformaldehyde for 10 min,

washed again with PBS, and then permeabilized with 0.25% Triton X-100 in PBS. This was subsequently followed with a further PBS wash and mounted with ProLong Gold Antifade Mountant with DAPI. The parasites were imaged using the same parameters in a NIKON Ni-E microscope Plus C2+ equipped with a DS-Ri2 Microscope Camera. The fluorescent foci and intensity were evaluated for vacuoles containing different numbers of parasites in the TIR1 and AID lines. To better compare the fluorescence in different lines, all vacuoles containing the same numbers of parasites (8 parasites/vacuole) on images were used for further analysis. The fluorescent foci in the vacuoles were counted automatically using NIKON microscopy software, and the intensities of the vacuoles were also measured automatically by the system. At least 50 vacuoles were scored for each replicate of three independent experiments with triplicates. The intensities of BODIPY-FL $C_{12}$ in vacuoles were measured in a similar way, and the total fluorescence in vacuoles were measured.

Salvage of cholesterol by the intracellular parasites was assessed by uptake of fluorescent TopFluor-Cholesterol (#810255P, Avanti Polar Lipids), following a protocol in previous studies[100,101]. In brief, parasites were grown in 500 μM auxin, or ethanol alone (0.1%) for 36 h on HFF monolayers on coverslips with DMEM supplemented with 2.5% Lipoprotein Depleted fetal bovine Serum (LPDS) (#880100-2, Kalen Biomedical). The TopFluor Cholesterol-LDL complex was prepared by incubation of 0.2 mg/ml human LDL (#L8292, Sigma-Aldrich) and 0.5 mg/ml TopFluor Cholesterol in DMEM supplemented with 2.5% LPDS at 37 °C overnight, to allow formation of the TopFluor Cholesterol-LDL complex. The parasites on monolayers were continued to be cultured in the DMEM containing the freshly prepared TopFluor Cholesterol-LDL complex for 30 min, washed with PBS twice, fixed by 4% paraformaldehyde and permeabilized by 0.25% Triton X-100 in PBS. The parasites in HFF cell monolayers were mounted in ProLong Gold Antifade Mountant with DAPI, followed by imaging under the green channel and DIC channel using a Nikon Ni-E microscope Plus C2 equipped with a DS-Ri2 Microscope Camera. The fluorescence in vacuoles containing different numbers of parasites were evaluated, and all vacuoles with 8 parasites on images were used for the measurement, to standardize the comparison of the fluorescence in different lines. The fluorescence for all the vacuoles (8 parasites) on images were measured automatically by the NIS elements AR 5.11.01.

### Fitness of extracellular parasites
Parasite membrane integrity was assayed using a live/dead cell imaging kit (Thermo Fisher Scientific, R37601), following the manufacturer recommended protocol. Parasites were grown in 500 μM auxin (IAA) or 0.1% ethanol alone for 36 h and harvested by mechanical passing through 22G needles and by filtration through 3.0 micron polycarbonate membranes. The freshly prepared parasites were resuspended in DMEM, at concentration of ~2 × 10⁷/ml, and then mixed with the same volume of mixture of vial A and vial B. The parasite mixtures were transferred to 24-well plates containing poly-lysine coated coverslips, centrifuged at $50 \times g$ for 1 min, followed by an incubation at 37 °C for 15 min. The parasites were washed with PBS, followed by fixation with 4% paraformaldehyde for 10 min. The coverslips were mounted for immediate visualization and imaging using a ×40 objective on a Nikon Ni-E confocal microscope. At least 100 parasites on each coverslip were counted, and the percentage of red parasites in the population was calculated and plotted. Two-three independent experiments with triplicates were performed for the assay.

### Immuno-staining for electron microscopy (Immuno-EM)
Parasites were grown on HFF monolayers for 24 h and, subsequently digested with trypsin for collecting samples. Samples were fixed in a 0.1 M sodium cacodylate buffer (pH 7.4) containing 3% (w/v) paraformaldehyde, 0.1% glutaraldehyde (v/v), 4% sucrose (w/v) at 4 °C

overnight. The samples were washed four times in a 0.1 M sodium cacodylate buffer containing 4% (w/v) sucrose, before neutralization on ice using a 0.1 M sodium cacodylate buffer containing 4% (w/v) sucrose and 0.1 M glycine, and subsequent dehydration in ethanol. The samples were immersed in LR White resin (Ted Pella, Inc.) at 4 °C overnight, subsequently polymerized at a light intensity of $1.2 \times 10^5$ μJ/cm² on ice for 5 h. The LR White imbedded samples were sectioned with a Leica ultramicrotome (EM UC7) at 100 nm and were picked onto 100-mesh formvar-carbon-coated nickel grids. The loaded nickel grids were then blocked and incubated with mouse anti-Ty antibodies (1:5) at RT for 2 h and overnight at 4 °C, followed with washing in PBS. The samples on grids were incubated with Alexa Fluor™ 488 goat anti-mouse IgG 10 nm colloidal gold (1:100) for 1 h, followed with washing in PBS and double-distilled water. The samples on grids were stained by 3% phosphotungstic acid, then observed using a JEM1400FLASH transmission electron microscope operation at 120 KV.

### Transmission electron microscopy (TEM) and electron tomography
Parasites were grown on HFF monolayers in auxin for 30 h, digested with trypsin for collecting samples, fixed using a fixative mixture of 2% paraformaldehyde and 3% glutaraldehyde at 4 °C overnight. The samples were washed and fixed again in 1% osmium tetroxide (Polysciences Inc., Warrington, PA) in 50 mM phosphate buffer at 4 °C for 1 h, washed three times and dehydrated by 30, 50, 70, 80 and 95% ethanol, and then by acetone, and then immersed in acetone and Spurr for 1 h. The samples were imbedded in Spurr and sectioned at 100 nm using a Leica ultramicrotome (EM UC7). For conventional TEM, the ultrathin-sections were collected on 100-mesh grids. For serial TEM, the ultrathin-sections were collected on Formvar-coated slot grids. The grids were then viewed by JEOL transmission electron microscopy (JEM1400FLASH). For electron tomography, the samples were sectioned at 260 nm by a Leica ultramicrotome. Sections were collected on formvar-coated slot grids and 10 nm colloidal gold fiducials were applied to the sections before acquiring single-axis tilt series with a tilt range of ± 48° with 3° increments around each axis under a Thermo-Scientific transmission electron microscope (Talos F200C G2) operation at 200 KV.

### RNA-seq and analysis
Parasites were grown in the presence of auxin (IAA) for 12 and 36 h. After washing with cold PBS, parasites were harvested by passing through 22G needles and filtration through 3.0 micron polycarbonate membranes. The parasites were immediately re-suspended in cold Trizol (Invitrogen), followed by total RNA extraction according to the manufacturer's protocol. RNA quality was monitored using an Agilent 2100 Bioanalyzer and only high-quality samples were used for constructing libraries and sequencing using Illumina HiSeq2500 by Gene Denovo Biotechnology Co. (Guangzhou, China). To acquire clean data for the following assembly and analysis, fastp (version 0.18.0) was firstly used to filter low quality bases and adapters, the software bowties2 (version 2.2.8) was used for mapping reads to the ribosomal RNA (rRNA) database to reduce the influence of the rRNA in the sample on the results. The remaining unmapped reads were aligned to the reference genome (ToxoDB release 53 https://toxodb.org/toxo/app) using HISAT2.2.1. The mapped reads of each sample were assembled by StringTie (version 1.3.1), and the expression was quantified by calculating the "fragments per kilobase of transcript per million mapped reads (FPKM)" value. The full transcriptome datasets were analyzed to identify the genes featuring a $p$ value below 0.05 and absolute fold change above 1.5, which were considered as differentially expressed genes (DEGs) between the groups. The DEGs were identified, with the "DEseq2" Package in R, and all DEGs among these groups were integrated to perform the Fuzzy C-means clustering analysis, using the "Mfuzz" Package in R, to explore the expression patterns during the

different developmental stages. The statistical analysis was performed in the software based on hypergeometric test.

## Weighted gene co-expression network analysis (WGNCA)

WGCNA was applied to the full transcriptome dataset (6770 genes; mean expression in all samples >1) commercially generated above, using the "WGCNA" package in R. A co-expression network was constructed with a beta value of 5 in the analysis. While genes were clustered into branches of highly expressed genes, featured modules were detected using the dynamic tree cutting method. The differential groups (the control vs. the AID lines at 12 and 36 h) were used as a trait for the WGCNA. the group-specific modules were identified based on the module-trait relationship (the correlation between eigengene and traits). Modules with a correlation coefficient >0.55 and $p$ value <0.05 were identified as a group-specific module. In addition, genes of these group-specific modules were further used for functional enrichment analysis using the ToxoDB database. The statistical analysis was performed in the software based on hypergeometric test.

## Untargeted metabolomics by GCMS

Parasites were grown in the presence of 500 μM auxin (IAA) for 36 h, subsequently washed with cold PBS twice, scraped and mechanically forced to egress by filtration through 3.0 micron polycarbonate membranes. The parasite harvesting was performed either at room temperature (experiment 1) or on ice (experiment 2). Parasites were centrifuged at $500 \times g$ at 4 °C (>$2 \times 10^8$ parasites), and the pellets were immediately quenched by addition of −40 °C methanol and sterile ddH$_2$O (400 μl methanol + 100 μl ddH$_2$O). The parasite resuspension (250 μl) was added with 250 μl 80% methanol, and sonicated on ice 1 min with 5 cycles. The mixture was vortexed and centrifuged to collect the supernatants. The supernatant (120 μl) was mixed with 5 μl 50 μg/ml L-corleucine, and dried under a nitrogen stream. The residue was reconstituted in 20 μl of 20 mg/ml methoxyamine hydrochloride in pyridine (containing 5 μg/ml n-alkanes standards), and the resulting mixture was then incubated at 37 °C for 90 min. The mixture was further mixed with 20 μl BSTFA (with 1% TMCS), and incubated at 70 °C for 60 min prior to GC-MS metabolomics analysis. The quality control (QC) sample was pooled from all samples, processed as the sample preparation. The GC-MS analysis was performed on Agilent7890A/5975C GC-MS system (Agilent Technologies Inc., CA, USA). An OPTIMA 5 MS Accent fused-silica capillary column was utilized to separate the derivatives. Helium was used as a carrier gas at a constant flow rate of 1 ml/min through the column. The injection volume was 1 μl and the solvent delay time was 5 min. The initial oven temperature was held at 60 °C for 1 min, ramped to 240 °C at a rate of 12 °C/min, to 320 °C at 40 °C/min, and finally held at 320 °C for 4 min. The temperatures of injector, transfer line, and electron impact ion source were set to 250 °C, 260 °C, and 230 °C, respectively. The electron ionization (EI) energy was 70 eV, and data was collected in a full scan mode (m/z 50-600).

The peak picking, alignment, deconvolution and further processing of resulting raw data were carried out according to the previously published protocols[102]. The final data was exported as a peak table file, including observations, variables (rt_mz), and peak areas. The data was normalized against the total peak value of total peaks, and differential metabolites were identified by statistical analysis performed in R platform. In the analysis, parametric and nonparametric tests were carried out using one-way ANOVA with Dunnett's multiple comparison, to identify differential metabolites, with $p < 0.05$ and a fold change >1.2, for further analysis. Enrichment of Metabolic Pathways were analyzed using the differential metabolite database commercially generated above. MetaboAnalyst 5.0 server was used to enrich the metabolite sets with KEGG, in which the $p$ value below 0.05 was considered to be significant. The statistical analysis was performed in the software based on hypergeometric test.

## Untargeted metabolomics by LCMS

The above parasite resuspension (250 μl) (80% methanol) was mixed with 250 μl 80% methanol, and sonicated on ice with 1 min on and 1 min off for 4 cycles, followed by incubation on ice for 30 min. After centrifugation at $15,000 \times g$, 4 °C and 15 min, the supernatant (200 μl) was dried under a nitrogen stream, and reconstituted in 40 μl of 10% methanol (including 100 ng/ml hexadecanoyl-L-carnitine-d$_3$, decanoyl-L-carnitine-d$_3$ and 1000 ng/ml valine-$^{13}C_5$-$^{15}$N, leucine-$^{13}C_6$, decanoic-d$_{19}$ acid, octadecanoic-d$_{35}$ acid, tetradecanoic-d$_{27}$ acid, phenylalanine-d$_5$, 3-chloro-D-phenylalanine and octanoic-d$_{15}$ acid) prior to performing UHPLC-HRMS/MS analysis. The quality control (QC) sample was pulling all the prepared samples together. The LC-MS analysis was performed on a Thermo Fisher Ultimate 3000 UHPLC system with a Waters ACQUITY UPLC BEH C18 column (2.1 mm × 100 mm, 1.7 μm), using standard positive and negative modes. For the positive mode: the mobile phases consisted of (A) water and (B) methanol, both with 0.1% formic acid. A linear gradient elution was performed with the following program: 0 min, 2%B; 12 min, 95%B; 15 min, 100%B and held to 18.1 min; 18.1 min, 2%B and held to 20 min. For the negative mode: the mobile phases contained (A) water and (B) methanol/water (95:5, v/v), both with 6.5 mM ammonium bicarbonate. A linear gradient elution was performed with the following program: 0 min, 2%B; 9 min, 70%B; 14 min, 100%B and held to 18 min; 18.1 min, 2%B and held to 20 min. The flow rate was 0.25 ml/min, and the injection volume was 2 μl for the positive mode and 3 μl for the negative mode.

The MS acquisition was performed on a Thermo Fisher Q Exactive Hybrid Quadrupole-Orbitrap Mass Spectrometry (QE) in Heated Electrospray Ionization Positive (HESI+) and Negative (HESI−) mode. The full scan was performed at a range of $m/z$ = 70–1000, with the resolution set at 35,000 and the AGC target at $3 \times 10^6$. The fragment ion of the top 10 precursors in each scan was acquired by Data-dependent acquisition (DDA) with HCD energy at 20, 30, and 40 eV, mass resolution of 17,500 FWHM and AGC Target of $5 \times 10^5$. The raw data was transformed to mzXML format by ProteoWizard and then processed by XCMS and CAMERA packages in the R software platform. The final data was exported as a peak table file, including sample names, variables (rt_mz) and peak areas. The peak areas data was normalized to internal standards before the statistical analysis. The in-house database was referenced to identify the compound by using $m/z$ (MS1), mass spectra (MS2) and retention times (RT). Normalization for relative parasite amounts was based on the total integrated peak area values of metabolites within an experimental batch. Similar analysis was performed for the identification of differential metabolites and enrichment of pathways as described for the GC-MS analysis.

## Scanning electron microscopy (SEM)

Parasites were grown in 500 μM auxin (IAA) for 36 h. After washing with cold PBS, parasites on HFF monolayers were mechanically forced to egress, and harvested by passing through 22 needles and filtration through 3.0 micron polycarbonate membranes. Parasite pellets were immediately fixed with 2.5% glutaraldehyde for 48 h in dark at 4 °C, and rinsed many times with PBS, followed by fixation in 1% osmium acid for 2 h. The samples were rinsed many times with PBS, dehydrated sequentially with 30, 50, 60, 70, 80, 90, and 100% ethanol for 15 min on each gradient and subsequently dried in a Leica CPD drying instrument (Leica, Weztlar, Germany). Samples were sprayed with an ion sputtering instrument (Hitachi MC1000, Hitachi, Japan), followed by observation in a Cold Field-Emission SEM (Hitachi SU-8010, Hitachi, Japan).

## Parasite virulence in mice

The parasite lines were used to challenge mice intraperitoneally with 100 parasites per mouse[76,77]. In brief, after parasite infection, the mice were randomly assigned to the IAA supplied group and the control group in separate cages with 5 mice in each. Mice were orally

administered with either auxin (3-indoleacetic acid, IAA, Sigma) or the vehicle by drinking water or by an intraperitoneal injection on a daily basis. The IAA supplied mouse group drunk sterile water containing 1 mg/ml IAA, 3 mM NaOH, 5% sucrose (w/v), flavored with 2 mg/ml TANG (Mondelēz International), pH 8.0, while the control group received the same drinking water but lacking IAA. The IAA supplied mouse group was intraperitoneally injected with a daily 0.2 ml of sterile solution containing 15 mg/ml IAA, 1 M NaOH, pH 7.8, while the control group received the same intraperitoneal injection but lacking IAA. The mice were monitored daily for bodyweight and health control. As the experiments did not deal with the infection related to sex, the mouse sex was not considered as a factor affecting the outcomes.

## Statistical analysis

All data were collected and analyzed blind. Data were analyzed using Prism software (version 8.0.2; Graphpad). Normally distributed data were tested using one-way or two-way ANOVA with Tukey's multiple comparison, while data with small sizes or abnormal distributions were tested using one-way ANOVA with Dunnett's multiple comparison. $p < 0.05$ was considered significant. Experiment-specific statistical information is provided in the figure legends or associated method details including replicates ($n$), trials ($N$), and statistical tests performed.

## Reporting summary

Further information on research design is available in the Nature Portfolio Reporting Summary linked to this article.

## Data availability

Due to the raw proteomics data are not available anymore, the scaffold file directly exported from the raw data has been deposited in the OMIX, China National Center for Bioinformation/Beijing Institute of Genomics, Chinese Academy of Sciences with accession number: OMIX001158. The metabolomics data have been deposited in the MetaboLights with accession number: MTBLS7006. Minimally processed data of the proteomics and metabolomics data are available in Supplementary Data 4, 8, 15 and 16, respectively. Transcriptomics sequencing data are available in the short read archive (SRA) of database of National Center for Biotechnology Information under the accession number of PRJNA812267. Minimally processed data are available in Supplementary Data 10 and 11. *Toxoplasma gondii* genome information can be found in ToxoDB release 53 (http://toxodb.org) and Eukaryotic pathogen, Vector & Host Information Resources can be found in VEupathDB (http://veupathdb.org). Source data are provided with this paper.

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

## Acknowledgements

We thank Prof. Tobias Spielmann and Prof. Nishith Gupta for their comments on this manuscript, Dr. Sophie Alvarez and Dr. Michael Naldrett at the Proteomics and Metabolomics Facility, Center for Biotechnology at the University of Nebraska-Lincoln for the proteomics analysis, Dr. Kevin Titeca for his assistance with SFINX graphics, Dr. Wen-Chao Wang at the Shanghai Profleader Biotech Co., Ltd for the untargeted metabolomics analysis, members in the EM facility lab at the School of Life Sciences, Sun Yat-sen University, and the GuangZhou Gene Denovo Biotechnology Company for the transcriptomic analysis, Dr. Jianming Zeng (University of Macau) and members of his bioinformatics team for sharing the biorstudio high performance computing

cluster (https://biorstudio.cloud) at Biotrainee in our transcriptomics analysis. We thank Prof. Vern Carruthers for the kind gift of RH*Δku80Δhxgprt* line, and Prof. Philippe Bastin for the BB2 hybridoma line for generation of monoclonal antibodies against the epitope tag Ty. This research was supported by the National Natural Science Foundation of China (31873009 to S.L. and 31772445 to D.H.L.), the National Key Research and Development Program of China (2021YFC2300802) and the Fund for Shanxi "1331 Project" (20211331-13) to X.Q.Z., and the University Startup Package to S.L.

## Author contributions

S.L. conceived the project. L.D.S., X.Q.Z., Z.R.L., Q.L. and G.H. provided insightful discussions and constructive suggestions. W.W., H.D., S.L. and X.T. designed and performed experiments. D.H.L. and J.Y. performed the immuno-EM and TEM. K.H. analyzed the datasets of proteomics, transcriptomics and metabolomics. S.L. supervised and interpreted the experimental work. S.L. wrote the paper with editorial input from L.D.S., G.H., D.H.L., Z.R.L. and X.Q.Z. All authors critically reviewed and approved the manuscript.

## Competing interests

The authors declare no competing interests.
