## [Peer Review File · Nature Communications]

Reviewer #1 (Remarks to the Author):

The manuscript by Wan and colleagues describes the identification and molecular dissection of several components that localize to the micropore of *Toxoplasma gondii*. The micropore has long been described by electron microscopy and hypothesized to be a site for endocytosis; however, no molecular markers have been described in *Toxoplasma* that would allow molecular dissection of the structures. The present study rigorously defines several new components that localize to these structures and attempts to examine their function through a series of phenotypic assays. Although the AID system employed for knockdown in principle allows for rapid knockdown (minutes) proteins are generally depleted for >12 h prior to phenotypic analysis. This raises the possibility of pleiotropic effects and, worse, the potential of comparing dead and living parasites. To support the functional characterization of the protein complexes, authors will have to examine the development of phenotypes as it corresponds to protein depletion and measure other facets of cellular viability. Separately, a major concern arises from several instances of image duplication encountered in the figures.

1. Several concerning instances of image duplication were encountered.
 - a. Fig. 1a top panel (EPS15-Ty/AP2u-HA) and Fig. 1b bottom panel (EPS15-Ty/MPP2-HA)
 - b. In Supplemental Fig. 3 panels for Ty-AID-PPG1 and EPS15-AID-HA are duplicated. Note additionally that the panels for UBP1-AID-6Ty are in the wrong orientation.
 - c. Fig. 2d EPS15-AID panels are the same as Supplemental Fig. 4a panels for EPS15-AID/AID-K13.
 - d. Supplemental Fig. 4a control (-IAA panels) for AID-AP2an and AP2d-AID are color-adjusted version of the same image.
2. Mention of gene ID's in the ToxoDB format in the main text would help with text mining and comparison to other studies. This could be accomplished in a table.
3. The morphology of the dying parasites should be better documented for the AID strains both to understand what is happening during depletion and to demonstrate that the different components of the complex share similar phenotypes.
4. It is surprising that salvage of such diverse molecules as host proteins, lipids, and biotin would all be affected by loss of the micropore. Given the dramatic effects on parasite replication associated with disruption of the micropore components, 36 h seems like an excessive depletion period before examining uptake of host proteins. The authors should select a single strain and follow the kinetics of IAA depletion and loss of host protein and biotin uptake. This would exclude the possibility that the parasites harvested after 36 h of depletion are simply dead. This concern is further heightened by the results of Supplemental Fig. 8b, which shows significant numbers of dead parasites following knockdown of the genes of interest.
5. There is no description of the viability and plaquing efficiency of the SPT mutants. Can the defect in host protein uptake be unambiguously attributed to endocytosis?
6. RNAseq analysis shows surprising variability in the response between strains with a large portion of the transcriptome changing. The authors should provide a clearer description of how the different mutants relate based on their transcriptomes and perform gene-ontology analysis on differences between mutants.
7. The metabolomic changes are similarly difficult to interpret. How do the authors explain differences in pyruvate, glutamic acid, and palmitelaidic acid between the knockdown of K13 alone and the triple knockdown? These inconsistencies in the data need to be more carefully considered.
8. The authors should ensure that error bars represent the variability between biological replicates. In some cases (e.g. Fig. 3C) the error appears too small to reflect biological replicates for that type of assay.

9. The overlay of so many strains in the survival graph makes it exceedingly difficult to follow individual lines. Authors are advised to separate the data into various graphs.

Reviewer #2 (Remarks to the Author):

The authors describe the molecular composition of the *T. gondii* micropore and the consequences of its disruption by genetic means on the parasite on several omics levels as well as microscopically and ultrastructurally. Using an impressive number of genetic constructs and mutant parasite cell lines they define the Kelch13 (K13) protein as the basic component of the endocytic micropore, identify other K13-interacting proteins and describe the perturbation of endocytosis upon induced knockdown of key components. While in the related parasite *Plasmodium falciparum* K13's role as a key part of the cytostome (the homologous structure in the malaria parasite to the micropore) has been described recently, the current work stands by itself and greatly expands our knowledge on this so far little studied structure of great importance for the *T. gondii* cell in nutrient acquisition and beyond. Overall, the conclusions are consistent with the presented experimental results and previous studies. The experimental approaches are state-of-the-art and reasonably well presented. However, numerous points have weaknesses that need to be addressed.

Major points

- The Discussion is somewhat short and disappointing and should be expanded. For instance, and not limited to these suggestions, K13 mutations have been shown to be involved in artemisinin resistance also in *T. gondii*, and a connection to mitochondrial metabolism has been provided (PMID: 32968076; 31806760; 33483501). Given the authors' metabolomics data it surprises that no mentioning and discussion of this aspect was done. Moreover, the prospects of the K13 KD for studying the role of K13 in artemisinin resistance by complementation with respective K13 mutants should be an aspect that can be discussed.
- Quite often sentences are hard to understand and it is sometimes guesswork what is meant. This makes me wonder whether the two native English speakers of the author's list (GH, LDS) have read the manuscript thoroughly enough. GH's contribution to the ms is not mentioned at all in the author contributions section! Please clarify contributions, and work extensively on the text.
- "Generation of *T. gondii* Lines": this chapter is hard to follow only by the information given in the tables, given the numerous different lines and constructs. At least the basic backbone of the non-commercial vectors should be provided as graphs, together with the insertion points of amplicons in the gene-of-interest. Moreover, not a single confirmative PCR for all the genomic insertions or KD is shown. While this might be considered ok for epitope or AID tagging, it does not exclude erroneous insertion at unintended loci. Please provide those PCRs at least for the most important parasite lines, and PCR primer locations added to the basic maps asked for above.
- Fig 4: the quenching temperature in the method parts is conflicting the quenching temp mentioned in the Fig legend (-40° / -20°C). Several metabolites such as GABA, lysine, fumaric acid and glutamate should be detectable on both MS-platforms. How did the authors integrate these results? The fold-regulation data shown should not cherry-pick from either one experiment but integrate data from both experiments.
- Line 802ff: The description of the experiment is insufficient. Please add LC solvent gradient information and more details on the MS method such as resolution, data dependent MS2 measurements, rapid switch between pos and neg modes etc. Please also add information on how the metabolites were identified and how the internal standards were used in analyzing both GCMS and LCMS experiments. The complete list of detected metabolites, including info on their ID status, should be made available to the reader.
- Fig 3f & g: I am not convinced how reliably this type of analysis can be done with such a noisy background staining. How many individual cells were analyzed for each mutant and time point? Nothing is given in M&M, and if it is done by human counting (3g) the images should be blinded to prevent bias. Was this done?
- Fig. 3b & d: the blot/IFA for TIR1 is missing. In addition, molecular weights of the detected proteins should be indicated in b. How many individual cells were analyzed for each mutant and time point?
- Sup. Fig 5d & e: How many individual cells were analyzed for each mutant and time point? In e

the blot for TIR1 is again missing. Please provide the source for the anti-LA antibody.

- Why is the triple kd less affected than the K13 kd in a number of metabolites (e.g. malate, glutamate, pyruvate, myristate and palmitelaidic acid)? Please comment.
- Line 349: please mention that the exp. was done on extracellular parasites. However, I don't understand the connection between the live/dead stain and the integrity of the micropore since this assay will detect any dead parasites, irrespective of the cause of death or absence or presence of a micropore. What is the rationale?
- Chapter "The micropore activity requires sphingoid bases": The rationale for the experiments is not immediate obvious, as are the conclusions. Please elaborate. Moreover, what is the connection between hemoglobin in Plasmodium and sphingoid bases in Toxo (line 341)?
- Suppl. Page 9 bottom line: should read "b-d". Also, in Suppl. Fig 5a, the parasite images are the same as in Fig. 3a – why? "Parasites with or without GFP accumulation (GFP+ and GFP-) were scored" – was this done blinded, and how was "negative" be defined? For instance, how did the GFP+ EPS15-AID Δ cpl parasites look like in comparison to GFP+ TIR1? How many cells were counted for each replicate?
- I don't understand what the authors mean by "The quantification showed defects of the IMC but no normal forms or other types in the key AID lines ... (Fig. 6a). Similarly: "SEM quantification showed the corrupted micropores correlated mostly with cases of material leakage in the key AID lines (Fig. 6d)." What material leakage are the authors talking about – there is nothing seen in the SEM images which was the basis for the quantification shown in 6d? Do I miss something? Please rephrase and clarify.
- No accession number to a metabolomic database is given. Please upload the raw and extracted data. The same is true for the raw RNAseq data!

Further comments

- Line 78: "T. gondii is the model organism for these parasites" – this makes little sense as the reference two sentences before is already T. gondii
- Line 135: please define "EH-, BAR-domain and AP2 homologs ". AP2 could be confused with the Apetala2 proteins of Tg.
- I recommend to make all images colorblind-friendly, in particular IFAs!
- Fig 1a,b: I suggest to replace the arrows with thinner lines. The arrow is confusing since it points to no object (directionality should be provided differently), and the thickness of the line obscures the objects that are connected.
- Sup Fig 1c: the signal with Strep488 not co-localizing with EPS15 should be explained/mentioned – I guess it's the apicoplast
- Line 166: "two AP2 subunits (i.e. AP2a and δ) were uniquely identified at the Toxoplasma micropore" – where is this shown?
- Line 181f. Seeing a budding vesicle in Fig. 1 g requires quite some imagination. I think the reader needs some graphical orientation (outline) in addition to the arrow
- Supplemental Video 1: please explain the colors
- Supplemental Fig. 2a. I'd like to see the distribution of domains in proteins which have more than one domain, similar to Fig 2b
- Line 192: It would be interesting to see whether the recently described fIDPnn algorithm's ability to predict putative propensities of disorder functions of IDPs (PMID: 34290238) is in accordance with this notion for the individual proteins
- Fig. 2b: why is the x-axis not simply scaled from 100 to 500?
- There is no named Suppl. table 8 but twice table 9! Please correct.
- A table summarizing the kd results after IAA treatment, in comparison to the phenotype score of the genome-wide CRISPR screen, would be informative
- Line 1217 Fig 4a: "and these modules were shown with their KEGG" – I don't understand what is meant.
- Fig S6: Please clarify whether metabolites shown in black were either not detected or not regulated.
- Sup Fig. 7: the plus/minus scheme is not in frame with the figure
- Line 328: define SPT1 and SPT2
- Fig.6c : stippled box for tKD is missing
- Line 436: have the authors searched ToxoDB for homologs of the Plasmodium cytosome constituents from Birnbaum et al. and which were not identified by BioID in this work? Such a comparison could be interesting.

- I didn't see a statement about sharing of materials (e.g. plasmids, cell lines). Please comment/add. Please also check whether all donations of reagents/cell lines from non-commercial sources are acknowledged.

Reviewer #3 (Remarks to the Author):

This paper describes the components of the *Toxoplasma gondii* micropore complex. This complex is composed of Kelch 13 and proteins involved in clathrin mediated endocytosis. The complex has features similar to the Plasmodium cytosome that has been characterized by several groups, suggesting that the functions and mechanics of the micropore are similar.

Very little has been understood about micropore function or the extent/importance of endocytosis in *T. gondii*. Thus this manuscript illuminates an important aspect of *T. gondii* biology that is likely to be preserved throughout the phylum. As discussed by the authors there are both conserved and unique features of the protein complex. A "hub" of this complex is Kelch 13, which has been linked to artemisinin resistance in malaria.

This study is a very comprehensive study that uses proteomics (BioID), genetics, cell biology, and advanced imaging. The study should be of interest to those interested in pathogenic eukaryotes as well, those interested in the evolution of the endocytic machinery, and those interested in the adaptations of parasitic organisms to unique intracellular environments. The important contribution of this manuscript is that it uses an "omics" approach to illustrate that the micropore is a likely site of endocytosis and that this activity is important for parasite nutrient acquisition from the host.

Throughout, the authors should state the numbers of parasites or vacuoles used for quantitation of uptake or fluorescence.

Figure 1-localization of micropore proteins. More details need to be provided about the number of parasites quantified, details of what was the input for the Pearson correlations, and statistical analysis (if done) of the images (IFA in panels a, b; EM in d, e).

Figure 2 Identification of Kelch 13 complex via biotin labeling/proteomics, and testing of essentiality using TIR regulated expression. Although I understand the point, I'm not sure figure 1b is particularly helpful. Panels d and e show important data—does it correlate with the CRISPR screen data?

(It may be the printer this reviewer used, but -/+ in panel e appear misaligned)

Figure 4 shows how the different mutants behave in RNA-seq and metabolite uptake. Supplemental figure 6 is more illuminating than panel 4a.

There is a lot of data in this paper and the authors use summary tables in the main body of the paper, but their points might be easier to understand if they instead highlighted a few key examples.

While this paper was in review, a complementary paper by the Waller group was uploaded to bioRxiv as a preprint and identifies the same protein complex. This preprint confirms many of the conclusions of this manuscript and should be viewed as complementary.
<https://doi.org/10.1101/2022.06.02.494549>.

RESPONSE TO REVIEWERS

*Author responses highlighted in blue

Reviewer #1 (Remarks to the Author):

The manuscript by Wan and colleagues describes the identification and molecular dissection of several components that localize to the micropore of *Toxoplasma gondii*. The micropore has long been described by electron microscopy and hypothesized to be a site for endocytosis; however, no molecular markers have been described in *Toxoplasma* that would allow molecular dissection of the structures. The present study rigorously defines several new components that localize to these structures and attempts to examine their function through a series of phenotypic assays. Although the AID system employed for knockdown in principle allows for rapid knockdown (minutes) proteins are generally depleted for >12 h prior to phenotypic analysis. This raises the possibility of pleiotropic effects and, worse, the potential of comparing dead and living parasites. To support the functional characterization of the protein complexes, authors will have to examine the development of phenotypes as it corresponds to protein depletion and measure other facets of cellular viability.

Response: We thank a lot the reviewer for her/his critical evaluation of our work, and we have provided further details about the development of phenotypes and parasite viability. Previously we had performed all the analyses, but did not include these in the first version of the manuscript, due to the reason of word limitation on the manuscript. We now summarize and present all these results in the main figures and supplementary figures. The GFP acquisition upon induction at 0, 12, 18, 24, 30 and 36 hrs showed that it significantly dropped and gradually decreased to a low level at 24 hrs and 30 hrs in the AID-Kelch13 line (Fig. 3a). The biotin levels basically mimicked this development of GFP acquisition (Supplementary Fig. 6c), suggesting that the phenotypes have developed gradually. However, our assays on the parasite viability (i.e. parasite egress and invasion) showed that the parasites could still actively egress from the vacuoles, although there is a delay in egress when stimulated by calcium ionophore treatment after auxin induction for 36 hours (Supplementary Fig. 7b). Unexpectedly, fresh extracellular parasites had the full capabilities to invade host cells after growing in auxin for 36 hours (Supplementary Fig. 7c). These results supported the fact that the parasites were viable at the induction point for the untargeted metabolomics and, therefore, that the metabolomics outcomes are not comparing dead with living parasites. We believe that our results, assayed with a

live/dead cell imaging kit, shows that fresh extracellular parasites (AID lines), while unstable, were not dead as they were still able to actively invade the host cells after growing in auxin for 36 hrs.

Furthermore, we functionally confirmed the defects found in the mitochondrial activities identified in the metabolomics. We did this by assaying parasite replication and mitochondrial membrane potential with parasites growing in regular media and media containing only one carbon source, such as glucose, glutamine and pyruvate. These results showed that the parasites depleted in the micropore proteins were not efficiently utilizing glucose to generate energy for the parasites by the mitochondrion (Supplementary Fig. 9). In summary, we believe that our functional characterizations are not associated with pleiotropic effects caused by auxin induction at 36 hs and at least, the phenotypic results we observed did not result from pleiotropic effects. We have included these results in the revised manuscript.

Separately, a major concern arises from several instances of image duplication encountered in the figures.

1. Several concerning instances of image duplication were encountered.
 - a. Fig. 1a top panel (EPS15-Ty/AP2u-HA) and Fig. 1b bottom panel (EPS15-Ty/MPP2-HA)
 - b. In Supplemental Fig. 3 panels for Ty-AID-PPG1 and EPS15-AID-HA are duplicated. Note additionally that the panels for UBP1-AID-6Ty are in the wrong orientation.
 - c. Fig. 2d EPS15-AID panels are the same as Supplemental Fig. 4a panels for EPS15-AID/AID-K13.
 - d. Supplemental Fig. 4a control (-IAA panels) for AID-AP2an and AP2d-AID are color-adjusted version of the same image.

Response: We thank so much the reviewer for pointing out the instances of image duplication. We have taken this problem very seriously by checking through all the original confocal images in our lab-owned microscope computer, and all the plaque plates and images. We have identified the mistakes at the stage of processing and uploading of the images, and we have now put the correct images with the current files. The raw images on IFA and plaque can be seen in the file of source data. We would like to state that it was not our intention to misuse the images – we apologise deeply for this error. This study has been complex and has been conducted over a long period of time, with the generation and identification of especially the AID lines in the supplementary figures. To confirm all the AID lines, we have presented the

results of the diagnostic PCRs in supplementary figure 3, and repeated plaque assays and obtained similar outcomes for all the AID lines in the supplementary figure 5 (new images used from the results of repeat experiments).

2. Mention of gene ID's in the ToxoDB format in the main text would help with text mining and comparison to other studies. This could be accomplished in a table.

Response: We thank very much the reviewer for the suggestion, and we have put the ID in the heatmap of the protein conservation in Fig. 2a, together with the CRISPR fitness scores of the genes. Also, we have compiled a Supplementary Table (Supplementary Table 5) on comparison of the proteins identified in this study (the *Toxoplasma* micropore) and the cytostome of *P. falciparum*

3. The morphology of the dying parasites should be better documented for the AID strains both to understand what is happening during depletion and to demonstrate that the different components of the complex share similar phenotypes.

Response: We have tested the cellular structures of intracellular parasites by light microscopy (Supplementary Fig. 7a), but we did not see obvious morphological changes though the micropore was corrupted when visualized by TEM and SEM, for parasites grown in auxin for 36 hours. On top of that, the parasites were still viable although their egress was delayed, since fresh extracellular parasites were fully active and able to invade host cells (Supplementary Fig. 7b-d). Besides, we have tested the stability of parasites, assayed by a live/dead cell imaging kit, for all the AID lines grown in auxin for 36 hrs and observed similar phenotypes for the growth-affecting AID lines (Supplementary Fig. 7e-f). The only exception was MPP1 but this was consistent with the results of the GFP acquisition (Supplementary Fig. 6a).

4. It is surprising that salvage of such diverse molecules as host proteins, lipids, and biotin would all be affected by loss of the micropore. Given the dramatic effects on parasite replication associated with disruption of the micropore components, 36 h seems like an excessive depletion period before examining uptake of host proteins. The authors should select a single strain and follow the kinetics of IAA depletion and loss of host protein and biotin uptake. This would exclude the possibility that the parasites harvested after 36 h of depletion are simply dead. This concern is further heightened by the results of Supplemental Fig. 8b, which shows significant numbers of dead parasites following knockdown of the genes of interest.

Response: We thank very much the reviewer for this suggestion, and we have provided the results on the GFP acquisition and biotin levels at the different induction time points (0, 12, 18, 24, 30, and 36 hrs)(Fig. 3b and Supplementary Fig. 6c). The kinetic induction showed that the GFP uptake dropped significantly at 12 hrs and approached a low level at 24-30 hours. However, our viability test suggested that the parasites were still viable at the induction time of 36 hrs (Supplementary Fig. 7b-c). Thereafter, the results of staining with a live/dead cell imaging kit did not necessarily indicate that the parasites depleted with the micropore proteins were dead, however, these parasites were clearly unstable and eventually lost the capabilities of host cell invasion (Supplementary Fig 7c-f). The live/dead cell imaging results should be resulted from leakiness of the parasite plasma membrane (PPM) at the micropore, which was supported by observation with TEM and SEM (Fig. 6).

5. There is no description of the viability and plaquing efficiency of the SPT mutants. Can the defect in host protein uptake be unambiguously attributed to endocytosis?

Response: We thank very much the reviewer for the comment, and we have provided the results of the plaque formation by the SPT mutant ($\Delta spt1\Delta spt2\Delta cpl$) that was used for the GFP uptake assay. However, the most recent study showed that the SPT mutant ($\Delta spt1\Delta spt2$) had clear growth defects, which were shown to be resulted from the defects of rhoptry secretion ¹. Therefore, it is uncertain how much extent the SPT enzymes contribute to parasite growth via the regulation of micropore endocytosis. Concerning what contributed to parasite growth via the micropore, we have discussed the metabolomics and nutrient acquisition in details in the discussion.

As to the second question, we observed reduced numbers of parasites with GFP foci, compared to the control line, and we did not see the obvious presence of parasites with diffused GFP fluorescence. The GFP diffusion indicates that the GFP is able to enter the parasites, but is unable to transport to the lysosome-like compartment (appears to be the GFP foci). Therefore, we concluded that this was an uptake defect. On the other hand, the uptake defect was observed in the SPT knockout in mammals and yeast ^{2,3}. However, we agree that we have no direct evidences to prove that the defect of GFP uptake was attributed to direct regulation on endocytosis by the early metabolites produced by SPT. Yet, these results suggested that maximal activity of the GFP uptake was associated with the ceramide *de novo* synthesis in parasites.

6. RNAseq analysis shows surprising variability in the response between strains with a large portion of the transcriptome changing. The authors should provide a clearer

description of how the different mutants relate based on their transcriptomes and perform gene-ontology analysis on differences between mutants.

Response: We thank the reviewer for this suggestion, and we have performed the GO analysis with differential transcripts by comparing the AID lines to the TIR1. However, we did not carry out the comparison of transcripts between the mutants. This is because we needed to consider the manuscript content and length which is already very large. Therefore, we think that the mutant difference should be left for future studies where we will be aiming to study the different functions of the micropore proteins.

7. The metabolomic changes are similarly difficult to interpret. How do the authors explain differences in pyruvate, glutamic acid, and palmitelaidic acid between the knockdown of K13 alone and the triple knockdown? These inconsistencies in the data need to be more carefully considered.

Response: We thank the reviewer for raising this question. However, we do not have a better explanation for the strain difference on the metabolomics. We considered that the differences may be resulted from depletion of additional proteins in the tKD line (i.e. EPS15 and PPG1), which may have other functions that are unknown.

8. The authors should ensure that error bars represent the variability between biological replicates. In some cases (e.g. Fig. 3C) the error appears too small to reflect biological replicates for that type of assay.

Response: Thanks a lot for the suggestion, we have repeated the analysis and exported the figure with error bars.

9. The overlay of so many strains in the survival graph makes it exceedingly difficult to follow individual lines. Authors are advised to separate the data into various graphs.

Response: We thank the reviewer for this suggestion again and we have re-organized the graph. We hope the current version displays the results more clearly (Fig. 7).

Reviewer #2 (Remarks to the Author):

The authors describe the molecular composition of the *T. gondii* micropore and the

consequences of its disruption by genetic means on the parasite on several omics levels as well as microscopically and ultrastructurally. Using an impressive number of genetic constructs and mutant parasite cell lines they define the Kelch13 (K13) protein as the basic component of the endocytic micropore, identify other K13-interacting proteins and describe the perturbation of endocytosis upon induced knockdown of key components. While in the related parasite *Plasmodium falciparum* K13's role as a key part of the cytostome (the homologous structure in the malaria parasite to the micropore) has been described recently, the current work stands by itself and greatly expands our knowledge on this so far little studied structure of great importance for the *T. gondii* cell in nutrient acquisition and beyond. Overall, the conclusions are consistent with the presented experimental results and previous studies. The experimental approaches are state-of-the-art and reasonably well presented. However, numerous points have weaknesses that need to be addressed.

Response: We appreciate with greatly the reviewer's positive evaluation of our work. We have carefully done all our best to dig out possible functions and structural features using various approaches. We have added additional results with the parasite viability and stability in intracellular and extracellular parasites upon depletion of the proteins, consistently supporting our discovery in the study.

Major points

- The Discussion is somewhat short and disappointing and should be expanded. For instance, and not limited to these suggestions, K13 mutations have been shown to be involved in artemisinin resistance also *in T. gondii*, and a connection to mitochondrial metabolism has been provided (PMID: 32968076; 31806760; 33483501). Given the authors' metabolomics data it surprises that no mentioning and discussion of this aspect was done. Moreover, the prospects of the K13 KD for studying the role of K13 in artemisinin resistance by complementation with respective K13 mutants should be an aspect that can be discussed.

Response: We thank so much the reviewer for this suggestion to improve the discussion. Due to manuscript length (word limitation), we have removed lots of content in the discussion from the first draft. In the current version, we have provided deeper discussion especially on the metabolomics and artemisinin resistance. Hopefully, the discussion in the current version has covered most of the important points in the study. Due to the word limitation, we could not expand more on this part. We thank the reviewer very much for the suggestion to improve the manuscript.

- Quite often sentences are hard to understand and it is sometimes guesswork what is meant. This makes me wonder whether the two native English speakers of the author's list (GH, LDS) have read the manuscript thoroughly enough. GH's contribution to the ms is not mentioned at all in the author contributions section! Please clarify contributions, and work extensively on the text.

Response: We thank the reviewer for the comments. In the current version, we have all made an effort to improve the quality of the text, and have made substantial revision on all sections. We apologise for the mistake on the contribution list, it was accidentally removed. The contribution list has been clarified.

- "Generation of *T. gondii* Lines": this chapter is hard to follow only by the information given in the tables, given the numerous different lines and constructs. At least the basic backbone of the non-commercial vectors should be provided as graphs, together with the insertion points of amplicons in the gene-of-interest. Moreover, not a single confirmative PCR for all the genomic insertions or KD is shown. While this might be considered ok for epitope or AID tagging, it does not exclude erroneous insertion at unintended loci. Please provide those PCRs at least for the most important parasite lines, and PCR primer locations added to the basic maps asked for above.

Response: We thank very much the reviewer for the comments. In the current version, we have made an illustration on the construction of the AID lines (Supplementary Fig. 3a-b), and made substantial revision to the text on the methods describing the generation of generic plasmids, amplicons and lines. We generated the tagging plasmids from published and addgene-stored plasmids that we had previously submitted. The revised version contains the details of the addgene resources and descriptions of the steps in generating the constructs. Furthermore, the generation of the fusion lines exactly followed the methods described in our previous paper where more detailed information was supplied. We have repeated the diagnostic PCRs and included the full PCR results in the Supplementary Fig 3.

- Fig 4: the quenching temperature in the method parts is conflicting the quenching temp mentioned in the Fig legend (-40° / -20°C). Several metabolites such as GABA, lysine, fumaric acid and glutamate should be detectable on both MS-platforms. How did the authors integrate these results? The fold-regulation data shown should not cherry-pick from either one experiment but integrate data from both experiments.

Response: We thank the reviewer for pointing out these problems. We have corrected the mistakes in the legend and methods. In the current version, we have provided results of LC and GC from both independent experiments in the main figure. As the two independent experiments were sampled at different temperatures (room temperature in Experiment 1 vs 4°C in Experiment 2), the outcomes of the metabolomics varied between experiments. We consider that the variations between experiments are mainly due to the sampling temperatures. In the former version, we considered it important not to waste resources on the measurements and included the outcomes of Experiment 1. However, we are not sure if it is necessary to present the data from experiment 1, or it is OK to present only the data from experiment 2? Though the outcomes of independent experiments varied, they were similar on most of the metabolites. We would be very interested to hear the suggestions from the reviewer.

- Line 802ff: The description of the experiment is insufficient. Please add LC solvent gradient information and more details on the MS method such as resolution, data dependent MS2 measurements, rapid switch between pos and neg modes etc. Please also add information on how the metabolites were identified and how the internal standards were used in analyzing both GCMS and LCMS experiments. The complete list of detected metabolites, including info on their ID status, should be made available to the reader.

Response: We thank the reviewer for the suggestions. We have revised the methods on the relevant contents, and provided the detailed information. The raw data of identified metabolites was minimally processed and provided in Supplementary Data 5 and Data 6.

- Fig 3f &g: I am not convinced how reliably this type of analysis can be done with such a noisy background staining. How many individual cells were analyzed for each mutant and time point? Nothing is given in M&M, and if it is done by human counting (3g) the images should be blinded to prevent bias. Was this done?

Response: We thank very much the reviewer for this question. We have updated the description of the method, which briefly described as followings. We evaluated the vacuoles in all the images from different lines, to ensure good quality of the pulsing assays and images (images were randomly taken under the same parameters in the microscope). To better compare the lines, all vacuoles containing 8 parasites on

images were used for further analysis. The fluorescent foci were counted automatically using NIKON microscopy software, and intensities of the vacuoles were also measured automatically by the system. At least 50 vacuoles were scored for each replica of three independent experiments with triplicates. The measurement should reflect the level of fluorescent reagent in the parasites. We believe that the processes of vacuole analysis should have assured the blindness.

- Fig. 3b & d: the blot/IFA for TIR1 is missing. In addition, molecular weights of the detected proteins should be indicated in b. How many individual cells were analyzed for each mutant and time point?

Response: We thank the reviewer for raising the question and are sorry that we did not mention the information in the legend. We have added all relevant information in the methods and legends to assure the proper manipulation of the study in the revised version.

- Sup. Fig 5d & e: How many individual cells were analyzed for each mutant and time point? In e the blot for TIR1 is again missing. Please provide the source for the anti-LA antibody.

Response: Sorry for not mentioning the information in the legend. We have updated the information. The fluorescence of vacuoles were measured automatically by the system as described for the BODIPY TR ceramide. At least 30 vacuoles were measured for each independent experiment and three independent experiments were performed.

- Why is the triple kd less affected than the K13 kd in a number of metabolites (e.g. malate, glutamate, pyruvate, myristate and palmitelaidic acid)? Please comment.

Response: We considered that it might be due to depletion of additional proteins in tKD (EPS15 and PPG1), and they may have other unknown functions. This was what we can think of.

- Line 349: please mention that the exp. was done on extracellular parasites. However, I don't understand the connection between the live/dead stain and the integrity of the micropore since this assay will detect any dead parasites, irrespective of the cause of death or absence or presence of a micropore. What is the rationale?

Response: We thank the reviewer for asking this question. Rationally, a probe in the kit detects live cells and stains these cells green, as Calcein AM can be catalyzed by an enzyme in live cells but not in dead cells. Instead, another probe, BOBO-3-iodide, stains DNA in dead cells, because it can permeabilize these cells, but not live cells. However, the kit detects not only cell viability, but also cell membrane integrity. Considering this information, we have re-organized our results and included new data in this current version. We tested parasite egress and invasion in parasites in auxin, and observed delayed egress defects, but they still retained the full capabilities to invade host cells with mechanically egressed and fresh parasites. However, these parasites dramatically lost their capabilities to invade host cells after incubation in DMEM for 3 hours. Based on these phenotypes, we proposed that the freshly egressed parasites depleted with the micropore proteins were still viable, but unstable. These results were likely the result of higher permeability at the micropore location, due to depletion of the micropore proteins. This point was supported by the results of TEM and SEM, which showed that the PPM at the micropore were corrupted.

- Chapter “The micropore activity requires sphingoid bases”: The rationale for the experiments is not immediate obvious, as are the conclusions. Please elaborate.

Response: We thank the reviewer for this suggestion. We agree with this, and we have revised this content, accordingly. As we have observed the effect on the GFP acquisition by deletion of the rate-limiting enzyme SPT, we considered that the effect should be due to the metabolites produced by SPT. SPT has been biochemically confirmed to produce sphingoid bases. However, as the reviewer pointed out that this was indirect evidence showing the effect by metabolites. So we have revised the statement, yet we hypothesize that the effect remains the same to the hemoglobin uptake in *P. falciparum*.

Moreover, what is the connection between hemoglobin in *Plasmodium* and sphingoid bases in Toxo (line 341)?

Response: We have deleted the sentences in the results, and we have placed it in the discussion, as followings. “The sphingoid bases act on the CME via the actin filaments in mammals and yeast ^{2,3}. In *Plasmodium* parasites actin plays a key role in hemoglobin uptake ^{4,5}. In addition, the *Plasmodium* SPT (PfSPT) has a very low fitness score (PlasmoDB), though the *de novo* synthesis seems not to be essential ⁶ and some of the genes, such as Des, is not identifiable in the genome. These analyses indicate that PfSPT may be involved in hemoglobin uptake”.

- Suppl. Page 9 bottom line: should read “b-d”. Also, in Supp. Fig 5a, the parasite images are the same as in Fig. 3a – why? “Parasites with or without GFP accumulation (GFP+ and GFP-) were scored” – was this done blinded, and how was “negative” be defined? For instance, how did the GFP+ EPS15-AID Δ cpl parasites look like in comparison to GFP+ TIR1? How many cells were counted for each replicate?

Response: We thank with greatly the reviewer for pointing out the mistakes on the expression, and we have updated the relevant information in the revised version. The images of parasites with or without GFP were just used as examples of the parasites with or without the fluorescence. We did not see parasites with diffused GFP fluorescence. Herein the GFP+ parasites denotes the parasites with GFP foci. What we performed on the counting was as follows. All parasites on images were counted together with the GFP containing parasites (GFP+). Those parasites without GFP fluorescence were considered as negative. In addition, we did not observe obvious differences on the appearances of parasites and GFP fluorescence between the +IAA AID line and +IAA TIR1. At least 100 parasites were counted for each replica, and three independent experiments with triplicates were performed.

- I don't understand what the authors mean by “The quantification showed defects of the IMC but no normal forms or other types in the key AID lines ... (Fig. 6a). Similarly: “SEM quantification showed the corrupted micropores correlated mostly with cases of material leakage in the key AID lines (Fig. 6d).” What material leakage are the authors talking about – there is nothing seen in the SEM images which was the basis for the quantification shown in 6d? Do I miss something? Please rephrase and clarify.

Response: We thank the reviewer for the suggestion on the writing. We have revised the sentences, as followings.

“The quantification assay showed that the normal micropore was unable to be found any more, instead only the defect forms were identified in the key AID lines (Fig. 6a).”
“We further examined the micropore in extracellular parasites using scanning electronic microscopy (SEM), and observed that the micropores were corrupted in the key AID lines in auxin, in many cases with a hanging membrane(Fig. 6c). SEM quantification clarified that all the micropores were corrupted, and most of them contained hanging membranes (Fig. 6d).”

We previously described the hanging membrane as material leakage, which was incorrect. Hopefully, the revised sentences are expressed correctly and correspond with the images.

- No accession number to a metabolomic database is given. Please upload the raw and extracted data. The same is true for the raw RNAseq data!

Response: We have uploaded the raw data to the OMIX (metabolomics) and NCBI (transcriptomics). The raw data were extracted and minimally processed for uploading in the Supplementary Data.

Further comments

- Line 78: “*T. gondii* is the model organism for these parasites” – this makes little sense as the reference two sentences before is already *T. gondii*.

Response: We thank for the reviewer for the suggestion. We have deleted the sentences.

- Line 135: please define “EH-, BAR-domain and AP2 homologs “. AP2 could be confused with the *Apetala2* proteins of *Tg*.

Response: We thank for the reviewer for the suggestion. We have provided relevant information in the sentence.

- I recommend to make all images colorblind-friendly, in particular IFAs!

Response: We have done this following the suggestion, and revised the IFA images in Fig. 1a and 1b.

- Fig 1a,b: I suggest to replace the arrows with thinner lines. The arrow is confusing since it points to no object (directionality should be provided differently), and the thickness of the line obscures the objects that are connected.

Response: We thank for the reviewer for this suggestion, and we have revised the arrows to thin lines.

- Sup Fig 1c: the signal with Strep488 not co-localizing with EPS15 should be explained/mentioned – I guess it’s the apicoplast.

Response: Yes, the additional spot is the apicoplast. We have added the information in the sentences. We thank the reviewer for the suggestion.

- Line 166: “two AP2 subunits (i.e. AP2a and σ) were uniquely identified at the *Toxoplasma* micropore” – where is this shown?

Response: We thank the reviewer for pointing out this misstatement. Here is the revised sentence “Moreover, two lineage specific proteins (i.e. MPP1 and MPP2, MicroPore Proteins) and another two AP2 subunits (i.e. AP2a and AP2 σ) were identified at the *Toxoplasma* micropore, while more additional proteins were identified at the cytostome of *P. falciparum* (Supplementary Table 5)”.

- Line 181f. Seeing a budding vesicle in Fig. 1 g requires quite some imagination. I think the reader needs some graphical orientation (outline) in addition to the arrow.

Response: We thank the review for this good suggestion. We have drawn a dotted line following the plasma membrane of the micropore at the first image (left), and placed this dotted line at each of the images. This way better allows readers to compare the images, and see the budding vesicle.

- Supplemental Video 1: please explain the colors.

Response: We have added the information to the video.

- Supplemental Fig. 2a. I'd like to see the distribution of domains in proteins which have more than one domain, similar to Fig 2b.

Response: We have added the distribution of domains in proteins (Supplementary Fig 2a).

- Line 192: It would be interesting to see whether the recently described fIDPnn algorithm's ability to predict putative propensities of disorder functions of IDPs (PMID: 34290238) is in accordance with this notion for the individual proteins.

Response: We thank so much the reviewer for this interesting suggestion, and we have performed the prediction, and mapped the IDP in proteins (Supplementary Fig. 2c).

- Fig. 2b: why is the x-axis not simply scaled from 100 to 500?

Response: We have revised the numbers in the Figure.

- There is no named Supp. table 8 but twice table 9! Please correct.

Response: Sorry for the mistakes. We have renumbered the tables, and carefully checked all the supplementary tables on texts.

- A table summarizing the kd results after IAA treatment, in comparison to the phenotype score of the genome-wide CRISPR screen, would be informative.

Response: We thank the reviewer for this suggestion. We have put the CRISPR information together with the conservation heatmap, and compiled a table to compare the components and functions for the *T. gondii* micropore and *P. falciparum* cytostome (Supplementary Table 5).

- Line 1217 Fig 4a: “and these modules were shown with their KEGG” – I don’t understand what is meant.

Response: We have revised the sentence as follows. “Three modules matched the requirements, and were then analyzed with enrichments of KEGG.”

- Fig S6: Please clarify whether metabolites shown in black were either not detected or not regulated.

Response: We have added the information in the legend.

- Sup Fig. 7: the plus/minus scheme is not in frame with the figure.

Response: We fell very sorry for the mistake, it might be caused by a figure printing process.

- Line 328: define SPT1 and SPT2.

Response: We have added the information.

- Fig.6c : stippled box for tKD is missing.

Response: We have added the information.

- Line 436: have the authors searched ToxoDB for homologs of the *Plasmodium* cytosome constituents from Birnbaum et al. and which were not identified by BioID in this work? Such a comparison could be interesting.

Response: We have compiled a table for the comparison on compositions and functions between the *Toxoplasma* micropore and *P.falciparum* cytosome (Supplementary Table 5).

- I didn't see a statement about sharing of materials (e.g. plasmids, cell lines). Please comment/add. Please also check whether all donations of reagents/cell lines from non-commercial sources are acknowledged.

Response: We thank very much the reviewer for the comments, and we have added the information under the section of Methods. The donations of reagents and cell lines were described in the Acknowledgements.

Reviewer #3 (Remarks to the Author):

This paper describes the components of the *Toxoplasma gondii* micropore complex. This complex is composed of Kelch 13 and proteins involved in clathrin mediated endocytosis. The complex has features similar to the Plasmodium cytosome that has been characterized by several groups, suggesting that the functions and mechanics of the micropore are similar.

Very little has been understood about micropore function or the extent/importance of endocytosis in *T. gondii*. Thus this manuscript illuminates an important aspect of *T. gondii* biology that is likely to be preserved throughout the phylum. As discussed by the authors there are both conserved and unique features of the protein complex. A "hub" of this complex is Kelch 13, which has been linked to artemisinin resistance in malaria.

This study is a very comprehensive study that uses proteomics (BioID), genetics, cell biology, and advanced imaging. The study should be of interest to those interested in

pathogenic eukaryotes as well, those interested in the evolution of the endocytic machinery, and those interested in the adaptations of parasitic organisms to unique intracellular environments. The important contribution of this manuscript is that it uses an "omics" approach to illustrate that the micropore is a likely site of endocytosis and that this activity is important for parasite nutrient acquisition from the host.

Response: We appreciate with greatly the reviewer's positive evaluation of our work. It was our aim to deeply dissect the function of the *Toxoplasma* micropore.

Throughout, the authors should state the numbers of parasites or vacuoles used for quantitation of uptake or fluorescence.

Response: We thank the reviewer for the suggestions, and we have gone through all our assays and added the information in the legends and methods.

Figure 1-localization of micropore proteins. More details need to be provided about the number of parasites quantified, details of what was the input for the Pearson correlations, and statistical analysis (if done) of the images (IFA in panels a, b; EM in d, e).

Response: We have provided the detailed information in the updated legend as suggested by the reviewer.

Figure 2 Identification of Kelch 13 complex via biotin labeling/proteomics, and testing of essentiality using TIR regulated expression. Although I understand the point, I'm not sure figure 1b is particularly helpful. Panels d and e show important data—does it correlate with the CRISPR screen data?

Response: We thank the reviewer for the comments. We think that Fig. 2b helps to understand the location of the micropore and the proximity of the proteins to the structures. We have provided the CRISPR information at the heatmap of protein conservation (Fig. 2a). The CRISPR fitness scores could help understand the essentiality of the proteins, but roughly matching the experimental data. We compiled a table on comparison of the *Toxoplasma* micropore and *P. falciparum* cystostome, and listed the essentiality and functions for the comparison (Supplementary Table 5).

(It may be the printer this reviewer used, but +/- in panel e appear misaligned).

Response: We thank the reviewer for reminding us of the problem, and we found the problem after submission. We have corrected it.

Figure 4 shows how the different mutants behave in RNA-seq and metabolite uptake. Supplemental figure 6 is more illuminating than panel 4a.

Response: We think so with the figures, yet we have not enough space to place it (Supplementary Fig. 6a). This is why it is in the supplementary figure.

There is a lot of data in this paper and the authors use summary tables in the main body of the paper, but their points might be easier to understand if they instead highlighted a few key examples.

Response: We thank the reviewer for the suggestion. To better understand the story and better compare the compositions and functions between the *Toxoplasma* micropore and the *P. falciparum* cytosome (Supplementary Table 5), we compiled a table that contains genes' accession numbers, essentiality and proteomics information. However, we are not sure that if it is necessary to place the table on the main body of the paper.

While this paper was in review, a complementary paper by the Waller group was uploaded to bioRxiv as a preprint and identifies the same protein complex. This preprint confirms many of the conclusions of this manuscript and should be viewed as complementary. <https://doi.org/10.1101/2022.06.02.494549>.

Response: We thank the reviewer for the information and comments. We have noticed the preprint. We are very happy to see this preprint.

** See Nature Portfolio's author and referees' website at www.nature.com/authors for information about policies, services and author benefits.

This email has been sent through the Springer Nature Tracking System
NY-610A-NPG&MTS

Confidentiality Statement:

This e-mail is confidential and subject to copyright. Any unauthorised use or disclosure of its contents is prohibited. If you have received this email in error please notify our Manuscript Tracking System Helpdesk team at <http://platformsupport.nature.com>.

Details of the confidentiality and pre-publicity policy may be found here <http://www.nature.com/authors/policies/confidentiality.html>

Privacy Policy | Update Profile

1. Nyonda, M.A. *et al.* Ceramide biosynthesis is critical for establishment of the intracellular niche of *Toxoplasma gondii*. *Cell Rep* **40**, 111224 (2022).
2. Zanolari, B. *et al.* Sphingoid base synthesis requirement for endocytosis in *Saccharomyces cerevisiae*. *EMBO J* **19**, 2824-33 (2000).
3. Meyer, S.G. *et al.* Myriocin, an inhibitor of serine palmitoyl transferase, impairs the uptake of transferrin and low-density lipoprotein in mammalian cells. *Arch Biochem Biophys* **526**, 60-8 (2012).
4. Smythe, W.A., Joiner, K.A. & Hoppe, H.C. Actin is required for endocytic trafficking in the malaria parasite *Plasmodium falciparum*. *Cell Microbiol* **10**, 452-64 (2008).
5. Lazarus, M.D., Schneider, T.G. & Taraschi, T.F. A new model for hemoglobin ingestion and transport by the human malaria parasite *Plasmodium falciparum*. *J Cell Sci* **121**, 1937-49 (2008).
6. Gerold, P. & Schwarz, R.T. Biosynthesis of glycosphingolipids de-novo by the human malaria parasite *Plasmodium falciparum*. *Mol Biochem Parasitol* **112**, 29-37 (2001).

Reviewer comments, second round -

Reviewer #1 (Remarks to the Author):

This is a manuscript from Wan et al. entitled "The Toxoplasma micropore mediates endocytosis for selective nutrient salvage from host cell compartments". It describes a protein complex that localizes at the micropore. Using knock-down strains for the different proteins identified, the authors show that these proteins are important for endocytosis. Mutant phenotypes were further explored using transcriptomics, metabolomics, and microscopy.

The accumulated data supports the conclusions. However, there are some critical points to address:

Major points:

- The transcriptomic data as presented in the manuscript is not easily understandable. It is unclear which comparisons are made. The pathways represented in Figure S8 appear hand-picked when looking at Table S9, S10, and S11 where the enrichment ratio is sometimes small. It is therefore unclear what is the biological significance of these enrichments. In figures 4 and S8, most of the apparent differences are apparent only at 36 hours of Auxin treatment and are most likely indirect consequences. As such, the authors should remove this data from the manuscript as it does not add any value to this well-crafted study. The reviewer did not have access to the transcriptomic data using the link provided in the manuscript.
 - The authors did not address correctly the R1 concern about the pleiotropic effects of the 36 hours of auxin treatment. The fact that parasites could invade cells after 36 hours of treatment merely indicates that they do not have any significant defect in their invasion apparatus, it does not necessarily mean that they are viable. Their inability to survive in the extracellular environment suggests that they are strongly affected by the depletion of these proteins. As pointed out by the R1, the reviewer would suggest reducing the auxin treatment to match the depletion of the proteins. However, the reviewer did not find in the manuscript or the supplementary figures an assessment of protein depletion by semi-quantitative western blot. This is a critical point since protein depletion cannot be assessed by IFA as shown in Figure S4. The AID system usually depletes target proteins within an hour but it may depend on their accessibility to the proteasome. This experiment should be performed. If the protein depletion is effective (within an hour), in line with R1, the reviewer would suggest repeating the experiment of GFP acquisition after short treatment of auxin (between 2 and 6 hours) to prove the direct impact of auxin treatment on the GFP acquisition.
 - R1 suggested better documenting the morphology of dying parasites. The authors only provide a single IFA of intracellular parasites in Figure S7a with IMC markers. Other organelles could be included such as the nucleus, Golgi and ER. Localization of the identified micropore component in the K13-AID line would also be of interest.
 - R1 pointed out image duplication, the reviewer is also concerned by some mishandling of data and statistics. For example, in Figure S7c the ESP15-AID strain + or - auxin show exactly the same data points. More concerning is the fact that the authors considered technical replicates (i. e. triplicates within the same biological replicate) as biological replicates. In all the graphs the data points are shown together, this could seriously skew the statistics as technical replicates are linked while biological replicates are not. For the quantification of the size of plaque assays, the authors only measured 9 plaques as stated in the legends of Figure 2f, while more than 100 plaques could have been scored. In figure 3h, data points are not shown in the graph.
- Additional minor points:
- The author state that "Kelch13 is essential for the growth of Toxoplasma in mice" which they did not measure. They only measured strain virulence but not their ability to propagate in mice. The reviewer would suggest changing the title.
 - The expression "extracellularly unstable" is unclear at best. It would be better to describe it as a "loss of fitness of extracellular parasites".

Reviewer #2 (Remarks to the Author):

Reviewer #2 (Remarks to the Author):

The authors describe the molecular composition of the *T. gondii* micropore and the consequences of its disruption by genetic means on the parasite on several omics levels as well as microscopically and ultrastructurally. Using an impressive number of genetic constructs and mutant parasite cell lines they define the Kelch13 (K13) protein as the basic component of the endocytic micropore, identify other K13-interacting proteins and describe the perturbation of endocytosis upon induced knockdown of key components. While in the related parasite *Plasmodium falciparum* K13's role as a key part of the cytostome (the homologous structure in the malaria parasite to the micropore) has been described recently, the current work stands by itself and greatly expands our knowledge on this so far little studied structure of great importance for the *T. gondii* cell in nutrient acquisition and beyond. Overall, the conclusions are consistent with the presented experimental results and previous studies. The experimental approaches are state-of-the-art and reasonably well presented. However, numerous points have weaknesses that need to be addressed.

Response: We appreciate with greatly the reviewer's positive evaluation of our work. We have carefully done all our best to dig out possible functions and structural features using various approaches. We have added additional results with the parasite viability and stability in intracellular and extracellular parasites upon depletion of the proteins, consistently supporting our discovery in the study.

Major points

- The Discussion is somewhat short and disappointing and should be expanded. For instance, and not limited to these suggestions, K13 mutations have been shown to be involved in artemisinin resistance also *in T. gondii*, and a connection to mitochondrial metabolism has been provided (PMID: 32968076; 31806760; 33483501). Given the authors' metabolomics data it surprises that no mentioning and discussion of this aspect was done. Moreover, the prospects of the K13 KD for studying the role of K13 in artemisinin resistance by complementation with respective K13 mutants should be an aspect that can be discussed.

Response: We thank so much the reviewer for this suggestion to improve the discussion. Due to manuscript length (word limitation), we have removed lots of content in the discussion from the first draft. In the current version, we have provided deeper discussion especially on the metabolomics and artemisinin resistance. Hopefully, the discussion in the current version has covered most of the important

points in the study. Due to the word limitation, we could not expand more on this part. We thank the reviewer very much for the suggestion to improve the manuscript.

ok

- Quite often sentences are hard to understand and it is sometimes guesswork what is meant. This makes me wonder whether the two native English speakers of the author's list (GH, LDS) have read the manuscript thoroughly enough. GH's contribution to the ms is not mentioned at all in the author contributions section! Please clarify contributions, and work extensively on the text.

For the most part ok, but there are still numerous typos, omissions, etc, in particular in the newly added text (one of many: "Resources and reagents may be directed to the corresponding author")!

Response: We thank the reviewer for the comments. In the current version, we have all made an effort to improve the quality of the text, and have made substantial revision on all sections. We apologise for the mistake on the contribution list, it was accidentally removed. The contribution list has been clarified.

- "Generation of *T. gondii* Lines": this chapter is hard to follow only by the information given in the tables, given the numerous different lines and constructs. At least the basic backbone of the non-commercial vectors should be provided as graphs, together with the insertion points of amplicons in the gene-of-interest. Moreover, not a single confirmative PCR for all the genomic insertions or KD is shown. While this might be considered ok for epitope or AID tagging, it does not exclude erroneous insertion at unintended loci. Please provide those PCRs at least for the most important parasite lines, and PCR primer locations added to the basic maps asked for above.

Response: We thank very much the reviewer for the comments. In the current version, we have made an illustration on the construction of the AID lines (Supplementary Fig. 3a-b), and made substantial revision to the text on the methods describing the generation of generic plasmids, amplicons and lines. We generated the tagging plasmids from published and addgene-stored plasmids that we had previously submitted. The revised version contains the details of the addgene resources and descriptions of the steps in generating the constructs. Furthermore, the generation of the fusion lines exactly followed the methods described in our previous paper where more detailed information was supplied. We have repeated the diagnostic PCRs and included the full PCR results in the Supplementary Fig 3.

For the most part ok, although the number of stars (*, indicating "unspecific bands" in the endo PCRs) is surprising and asks for an explanation.

• Fig 4: the quenching temperature in the method parts is conflicting the quenching temp mentioned in the Fig legend (-40° / -20°C). Several metabolites such as GABA, lysine, fumaric acid and glutamate should be detectable on both MS-platforms. How did the authors integrate these results? The fold-regulation data shown should not cherry-pick from either one experiment but integrate data from both experiments.

Response: We thank the reviewer for pointing out these problems. We have corrected the mistakes in the legend and methods. In the current version, we have provided results of LC and GC from both independent experiments in the main figure. As the two independent experiments were sampled at different temperatures (room temperature in Experiment 1 vs 4°C in Experiment 2), the outcomes of the metabolomics varied between experiments. We consider that the variations between experiments are mainly due to the sampling temperatures. In the former version, we considered it important not to waste resources on the measurements and included the outcomes of Experiment 1. However, we are not sure if it is necessary to present the data from experiment 1, or it is OK to present only the data from experiment 2? Though the outcomes of independent experiments varied, they were similar on most of the metabolites. We would be very interested to hear the suggestions from the reviewer.

It lies in the nature of experimental sciences that each experiment has to be performed multiple times, but at least twice, to distinguish robust from spurious data. In this respect it is very unfortunate that the two experiments differ by this drastic temperature difference during sampling. However, what is “N= “ referring to in the legend to Fig 4b (“Metabolic enrichment with differential metabolites identified in untargeted metabolomics from Experiment 1 (N=5) and Experiment 2 (N=3) (including LCMS and GCMS) for TIR1, AID-K13 and tKD after 36 hours of auxin induction.”)? Different numbers of cultures treated at different days but measured at one day? Also, what do you mean by “The empty boxes indicated the metabolites were not identified in experiment 1.”? There are empty green boxes and empty red boxes (as opposed to filled red boxes), but both, the green and red boxes contain data points. Please clarify. Depending on the answers it might be ok to present the two experiments as such (b-d).

• Line 802ff: The description of the experiment is insufficient. Please add LC solvent gradient information and more details on the MS method such as resolution, data dependent MS2 measurements, rapid switch between pos and neg modes etc. Please also add information on how the metabolites were identified and how the

internal standards were used in analyzing both GCMS and LCMS experiments. The complete list of detected metabolites, including info on their ID status, should be made available to the reader.

Response: We thank the reviewer for the suggestions. We have revised the methods on the relevant contents, and provided the detailed information. The raw data of identified metabolites was minimally processed and provided in Supplementary Data 5 and Data 6.

ok

• Fig 3f &g: I am not convinced how reliably this type of analysis can be done with such a noisy background staining. How many individual cells were analyzed for each mutant and time point? Nothing is given in M&M, and if it is done by human counting (3g) the images should be blinded to prevent bias. Was this done?

Response: We thank very much the reviewer for this question. We have updated the description of the method, which briefly described as followings. We evaluated the vacuoles in all the images from different lines, to ensure good quality of the pulsing assays and images (images were randomly taken under the same parameters in the microscope). To better compare the lines, all vacuoles containing 8 parasites on images were used for further analysis. The fluorescent foci were counted automatically using NIKON microscopy software, and intensities of the vacuoles were also measured automatically by the system. At least 50 vacuoles were scored for each replica of three independent experiments with triplicates. The measurement should reflect the level of fluorescent reagent in the parasites. We believe that the processes of vacuole analysis should have assured the blindness.

ok

• Fig. 3b & d: the blot/IFA for TIR1 is missing. In addition, molecular weights of the detected proteins should be indicated in b. How many individual cells were analyzed for each mutant and time point?

Response: We thank the reviewer for raising the question and are sorry that we did not mention the information in the legend. We have added all relevant information in the methods and legends to assure the proper manipulation of the study in the revised version.

ok

• Sup. Fig 5d & e: How many individual cells were analyzed for each mutant and time point? In e the blot for TIR1 is again missing. Please provide the source for the anti-LA antibody.

Response: Sorry for not mentioning the information in the legend. We have updated the information. The fluorescence of vacuoles were measured automatically by the system as described for the BODIPY TR ceramide. At least 30 vacuoles were measured for each independent experiment and three independent experiments were performed.

ok

• Why is the triple kd less affected than the K13 kd in a number of metabolites (e.g. malate, glutamate, pyruvate, myristate and palmitelaidic acid)? Please comment.

Response: We considered that it might be due to depletion of additional proteins in tKD (EPS15 and PPG1), and they may have other unknown functions. This was what we can think of.

Given the problems with the metabolomics data (see above; Fig 4) I need to see those answers before I can agree.

• Line 349: please mention that the exp. was done on extracellular parasites. However, I don't understand the connection between the live/dead stain and the integrity of the micropore since this assay will detect any dead parasites, irrespective of the cause of death or absence or presence of a micropore. What is the rationale?

Response: We thank the reviewer for asking this question. Rationally, a probe in the kit detects live cells and stains these cells green, as Calcein AM can be catalyzed by an enzyme in live cells but not in dead cells. Instead, another probe, BOBO-3-iodide, stains DNA in dead cells, because it can permeabilize these cells, but not live cells. However, the kit detects not only cell viability, but also cell membrane integrity. Considering this information, we have re-organized our results and included new data in this current version. We tested parasite egress and invasion in parasites in auxin, and observed delayed egress defects, but they still retained the full capabilities to invade host cells with mechanically egressed and fresh parasites. However, these parasites dramatically lost their capabilities to invade host cells after incubation in DMEM for 3 hours. Based on these phenotypes, we proposed that the freshly egressed parasites depleted with the micropore proteins were still viable, but unstable. These results were likely the result of higher permeability at the micropore location, due to depletion of the micropore proteins. This point was supported by the results of TEM and SEM, which showed that the PPM at the micropore were corrupted.

ok

• Chapter "The micropore activity requires sphingoid bases": The rationale for the experiments is not immediate obvious, as are the conclusions. Please elaborate.

Response: We thank the reviewer for this suggestion. We agree with this, and we have revised this content, accordingly. As we have observed the effect on the GFP acquisition by deletion of the rate-limiting enzyme SPT, we considered that the effect should be due to the metabolites produced by SPT. SPT has been biochemically confirmed to produce sphingoid bases. However, as the reviewer pointed out that this was indirect evidence showing the effect by metabolites. So we have revised the statement, yet we hypothesize that the effect remains the same to the hemoglobin uptake in *P. falciparum*.

Ok

Moreover, what is the connection between hemoglobin in *Plasmodium* and sphingoid bases in Toxo (line 341)?

Response: We have deleted the sentences in the results, and we have placed it in the discussion, as followings. “The sphingoid bases act on the CME via the actin filaments in mammals and yeast ^{2,3}. In *Plasmodium* parasites actin plays a key role in hemoglobin uptake ^{4,5}. In addition, the *Plasmodium* SPT (PfSPT) has a very low fitness score (PlasmoDB), though the *de novo* synthesis seems not to be essential ⁶ and some of the genes, such as Des, is not identifiable in the genome. These analyses indicate that PfSPT may be involved in hemoglobin uptake”.

Ok

• Suppl. Page 9 bottom line: should read “b-d”. Also, in Supp. Fig 5a, the parasite images are the same as in Fig. 3a – why? “Parasites with or without GFP accumulation (GFP+ and GFP-) were scored” – was this done blinded, and how was “negative” be defined? For instance, how did the GFP+ EPS15-AID Δ cpl parasites look like in comparison to GFP+ TIR1? How many cells were counted for each replicate?

Response: We thank with greatly the reviewer for pointing out the mistakes on the expression, and we have updated the relevant information in the revised version. The images of parasites with or without GFP were just used as examples of the parasites with or without the fluorescence. We did not see parasites with diffused GFP fluorescence. Herein the GFP+ parasites denotes the parasites with GFP foci. What we performed on the counting was as follows. All parasites on images were counted together with the GFP containing parasites (GFP+). Those parasites without GFP fluorescence were considered as negative. In addition, we did not observe obvious differences on the appearances of parasites and GFP fluorescence between the

+IAA AID line and +IAA TIR1. At least 100 parasites were counted for each replica, and three independent experiments with triplicates were performed.

ok

• I don't understand what the authors mean by "The quantification showed defects of the IMC but no normal forms or other types in the key AID lines ... (Fig. 6a).

Similarly: "SEM quantification showed the corrupted micropores correlated mostly with cases of material leakage in the key AID lines (Fig. 6d)." What material leakage are the authors talking about – there is nothing seen in the SEM images which was the basis for the quantification shown in 6d? Do I miss something? Please rephrase and clarify.

Response: We thank the reviewer for the suggestion on the writing. We have revised the sentences, as followings.

"The quantification assay showed that the normal micropore was unable to be found any more, instead only the defect forms were identified in the key AID lines (Fig. 6a)."

"We further examined the micropore in extracellular parasites using scanning electronic microscopy (SEM), and observed that the micropores were corrupted in the key AID lines in auxin, in many cases with a hanging membrane(Fig. 6c). SEM quantification clarified that all the micropores were corrupted, and most of them contained hanging membranes (Fig. 6d)."

We previously described the hanging membrane as material leakage, which was incorrect. Hopefully, the revised sentences are expressed correctly and correspond with the images.

ok

• No accession number to a metabolomic database is given. Please upload the raw and extracted data. The same is true for the raw RNAseq data!

Response: We have uploaded the raw data to the OMIX (metabolomics) and NCBI (transcriptomics). The raw data were extracted and minimally processed for uploading in the Supplementary Data.

ok

Further comments

• Line 78: "T. gondii is the model organism for these parasites" – this makes little sense as the reference two sentences before is already *T. gondii*.

Response: We thank for the reviewer for the suggestion. We have deleted the sentences.

ok

- Line 135: please define “EH-, BAR-domain and AP2 homologs “. AP2 could be confused with the Apetala2 proteins of Tg.

Response: We thank for the reviewer for the suggestion. We have provided relevant information in the sentence.

ok

- I recommend to make all images colorblind-friendly, in particular IFAs!

Response: We have done this following the suggestion, and revised the IFA images in Fig. 1a and 1b.

Why only those and not the others?

- Fig 1a,b: I suggest to replace the arrows with thinner lines. The arrow is confusing since it points to no object (directionality should be provided differently), and the thickness of the line obscures the objects that are connected.

Response: We thank for the reviewer for this suggestion, and we have revised the arrows to thin lines.

ok

- Sup Fig 1c: the signal with Strep488 not co-localizing with EPS15 should be explained/mentioned – I guess it's the apicoplast.

Response: Yes, the additional spot is the apicoplast. We have added the information in the sentences. We thank the reviewer for the suggestion.

ok

- Line 166: “two AP2 subunits (i.e. AP2a and σ) were uniquely identified at the *Toxoplasma* micropore” – where is this shown?

Response: We thank the reviewer for pointing out this misstatement. Here is the revised sentence “Moreover, two lineage specific proteins (i.e. MPP1 and MPP2, MicroPore Proteins) and another two AP2 subunits (i.e. AP2a and AP2 σ) were identified at the *Toxoplasma* micropore, while more additional proteins were identified at the cytostome of *P. falciparum* (Supplementary Table 5)”.

ok

- Line 181f. Seeing a budding vesicle in Fig. 1 g requires quite some imagination. I think the reader needs some graphical orientation (outline) in addition to the arrow.

Response: We thank the review for this good suggestion. We have drawn a dotted line following the plasma membrane of the micropore at the first image (left), and placed this dotted line at each of the images. This way better allows readers to compare the images, and see the budding vesicle.

ok

- Supplemental Video 1: please explain the colors.

Response: We have added the information to the video.

ok

- Supplemental Fig. 2a. I'd like to see the distribution of domains in proteins which have more than one domain, similar to Fig 2b.

Response: We have added the distribution of domains in proteins (Supplementary Fig 2a).

ok

- Line 192: It would be interesting to see whether the recently described fIDPnn algorithm's ability to predict putative propensities of disorder functions of IDPs (PMID: 34290238) is in accordance with this notion for the individual proteins.

Response: We thank so much the reviewer for this interesting suggestion, and we have performed the prediction, and mapped the IDP in proteins (Supplementary Fig. 2c).

ok

- Fig. 2b: why is the x-axis not simply scaled from 100 to 500?

Response: We have revised the numbers in the Figure.

ok

- There is no named Supp. table 8 but twice table 9! Please correct.

Response: Sorry for the mistakes. We have renumbered the tables, and carefully checked all the supplementary tables on texts.

ok

- A table summarizing the kd results after IAA treatment, in comparison to the phenotype score of the genome-wide CRISPR screen, would be informative.

Response: We thank the reviewer for this suggestion. We have put the CRISPR information together with the conservation heatmap, and compiled a table to compare the components and functions for the *T. gondii* micropore and *P. falciparum* cytostome (Supplementary Table 5).

ok

• Line 1217 Fig 4a: “and these modules were shown with their KEGG” – I don’t understand what is meant.

Response: We have revised the sentence as follows. “Three modules matched the requirements, and were then analyzed with enrichments of KEGG.”

ok

• Fig S6: Please clarify whether metabolites shown in black were either not detected or not regulated.

Response: We have added the information in the legend.

ok

• Sup Fig. 7: the plus/minus scheme is not in frame with the figure.

Response: We fell very sorry for the mistake, it might be caused by a figure printing process.

ok

• Line 328: define SPT1 and SPT2.

Response: We have added the information.

ok

• Fig.6c : stippled box for tKD is missing.

Response: We have added the information.

ok

• Line 436: have the authors searched ToxoDB for homologs of the *Plasmodium* cytosome constituents from Birnbaum et al. and which were not identified by BioID in this work? Such a comparison could be interesting.

Response: We have compiled a table for the comparison on compositions and functions between the *Toxoplasma micropore* and *P.falciparum* cytosome (Supplementary Table 5).

ok

• I didn’t see a statement about sharing of materials (e.g. plasmids, cell lines). Please comment/add. Please also check whether all donations of reagents/cell lines from non-commercial sources are acknowledged.

Response: We thank very much the reviewer for the comments, and we have added the information under the section of Methods. The donations of reagents and cell lines

were described in the Acknowledgements.

ok

REVIEWER COMMENTS

Reviewer #1 (Remarks to the Author):

This is a manuscript from Wan et al. entitled “The Toxoplasma micropore mediates endocytosis for selective³ nutrient salvage from host cell compartments”. It describes a protein complex that localizes at the micropore. Using knock-down strains for the different proteins identified, the authors show that these proteins are important for endocytosis. Mutant phenotypes were further explored using transcriptomics, metabolomics, and microscopy.

The accumulated data supports the conclusions. However, there are some critical points to address:

Response: We greatly appreciate the reviewer’s positive evaluation of our work. We thank the reviewer very much for his/her critical comments and we have done our best to revise and improve the manuscript. Below is the point-by-point response.

Major points:

- The transcriptomic data as presented in the manuscript is not easily understandable. It is unclear which comparisons are made. The pathways represented in Figure S8 appear hand-picked when looking at Table S9, S10, and S11 where the enrichment ratio is sometimes small. It is therefore unclear what is the biological significance of these enrichments. In figures 4 and S8, most of the apparent differences are apparent only at 36 hours of Auxin treatment and are most likely indirect consequences. As such, the authors should remove this data from the manuscript as it does not add any value to this well-crafted study. The reviewer did not have access to the transcriptomic data using the link provided in the manuscript.

Response: We thank the reviewer for his/her comments on the transcriptomics data. To address the relevant questions, we raise some points to explain why we performed the RNA-Seq analysis, how we analyzed the datasets and what the analysis told us about the micropore.

Firstly, *T. gondii* faces a complex interface between the parasite and the host cell and it is very difficult to find out what exact nutrients would be ingested via the micropore,

although we have clear results on the ingestion of host cytosolic GFP and biotin. To gain a broad view of the impact in the parasite upon depletion of the micropore proteins, we analyzed the -omics. It is relatively easier to identify differences by transcriptomics, in comparison with metabolomics. Therefore, we considered that the transcriptomics would bridge the detailed measurements (e.g. biotin, GFP) and the metabolomics, telling us about the overall changes in parasites with depleted micropore proteins.

Secondly, the transcriptomics appeared complex, especially when presenting the results by comparing each of the induction points to the control TIR1. In this case, we did not use that approach, instead we took advantage of the trend clustering (Mfuzz) and co-expression network (WGCNA) analyses, to observe overall changes by comparing 6 of the experimental groups (EPS15-AID, AID-K13 and tKD lines at two induction points) to the control TIR1. This way enabled us to find out the trends or modules that have similar expression features. These analyses have enriched our understanding of specific functions/pathways in the overall observation. Here we would like to point out that the modules in Fig. 4a, featured high or low expression of genes in the control TIR1 relative to 6 of the experimental groups, thus reflecting the overall changes in the parasites with depleted micropore proteins.

Thirdly, these analyses revealed changes of metabolic pathways, as shown in the module enrichment (Fig. 4a) and trend enrichment (Supplementary Fig. 9a). These results have thus provided us with clues that allow better analysis of the metabolomics datasets and a better presentation of the results.

As such, we suggest retaining the transcriptomics analysis in the manuscript. The overall analyses from the transcriptomics have led us to come to the conclusion that metabolic changes occurred in the parasites with depleted key micropore proteins (Fig. 4a and Supplementary Fig. 9a). From this angle, the hand-picked pathways are representatives of our overall analysis.

In addition, the overall analyses did not support the point that most of the differences are apparent only at 36 hours of auxin treatment. Closer observation of the data at the 36 hours did not show stronger differences at 36 hours in comparison with 12 hours in Fig. 4a, and Supplementary Fig. 9a. In most of the trends and modules, what we actually observed is steady (or almost steady) levels, such as seen with

modules 1-3 (Fig. 4a) and the trend clusters 2-4 (Supplementary Fig. 9a). The obvious case showing a gradual drop or increase is only the trend cluster 6 (Supplementary Fig. 9a). Overall, our data does not support the notion that the transcriptomics differences are a result of indirect consequences.

However, after careful checking, we have identified some mislabeling in the excel spreadsheets, which might have caused difficulties in understanding the transcriptomics. In addition, we agree that the description of the transcriptomics did not clearly point out the key information on the data analysis in the last version.

We thank the reviewer very much for the comments and suggestions, which have made us re-organize and re-present the key information about the transcriptomics analysis in the current version of manuscript. We have accordingly revised the description in the results and Legends.

- The authors did not address correctly the R1 concern about the pleiotropic effects of the 36 hours of auxin treatment. The fact that parasites could invade cells after 36 hours of treatment merely indicates that they do not have any significant defect in their invasion apparatus, it does not necessarily mean that they are viable. Their inability to survive in the extracellular environment suggests that they are strongly affected by the depletion of these proteins. As pointed out by the R1, the reviewer would suggest reducing the auxin treatment to match the depletion of the proteins. However, **the reviewer did not find in the manuscript or the supplementary figures an assessment of protein depletion by semi-quantitative western blot**. This is a critical point since protein depletion cannot be assessed by IFA as shown in Figure S4. The AID system usually depletes target proteins within an hour but it may depend on their accessibility to the proteasome. This experiment should be performed. If the protein depletion is effective (within an hour), in line with R1, the reviewer would **suggest repeating the experiment of GFP acquisition after short treatment of auxin (between 2 and 6 hours) to prove the direct impact of auxin treatment on the GFP acquisition**.

Response: We thank the reviewer for their helpful comments and suggestions. We have carried out the Western blots using quantitative measurements with AID-K13 in auxin for 0-6 hours (Fig. 3c). We have also repeated the GFP acquisition assay using the AID-K13 but not the AID-K13/ Δ cpl at the above induction time points. Previously,

we measured the GFP acquisition in the AID-K13/ Δ cpl by growth of the parasites in GFP-expressing host cells for 36 hours and by induction at auxin for 0, 12, 24 and 36 hours. From analysis of the AID-K13/ Δ cpl line, we observed a significant decrease at the 12 hour induction point. However, we need to point out that the host derived GFP has accumulated in the parasites during the initial 24 hours' growth, resulting in the outcome of the 12 hours' induction as being the combined effect of GFP degradation and GFP acquisition in the parasites. In the present assay, we turned to using the AID-K13 line where the cysteine protease CPL was inhibited by LHVS (10 μ M), at the induction initiation time point, as illustrated in Fig. 3d. This allows a better examination of the direct impact on the GFP acquisition during the conditional down-regulation of K13. This assay demonstrated reduced GFP acquisition at the induction points at 2, 4 and 6 hours (Fig. 3e), thus confirming the direct role of the micropore proteins on the GFP acquisition from host cell cytosol.

- R1 suggested better documenting the morphology of dying parasites. The authors only provide a single IFA of intracellular parasites in Figure S7a with IMC markers. **Other organelles could be included such as the nucleus, Golgi and ER.** Localization of the identified micropore component in the K13-AID line would also be of interest.

Response: We thank reviewer for these helpful comments and suggestions. We have repeated the IFA analysis in the AID-K13 parasites using antibodies against the IMC markers (GAP45, IMC1 and MLC1), the mitochondrial marker (Hsp60) and the apicoplast marker (ACP) (see Supplementary Fig. 7). At present, we have no antibodies against markers of the nucleus, Golgi and ER. However, we did not observe obvious changes to the structures analyzed on the IFA staining. The results were consistent with those using ACC1-HA (the apicoplast) and PC-HA (the mitochondrion) in the AID-K13 depleted parasites (Supplementary Fig. 6d). Instead, the parasite rosette orientation appeared to be lost, which was observed by differential interference contrast as well as the IFA staining (Supplementary Fig. 7). We hope that this further analysis has provided additional information on the morphology of the parasites after auxin induction.

In addition, the reviewer raised the suggestion that we should examine other identified micropore proteins in the AID-K13 line by genetic tagging and IFA. Considering that the current version is already a data-dense manuscript, we suggest

that this part of work is left for a future study. In the study, we will be aiming to make further studies on the protein interactions in the parasites, association of the micropore proteins with mitochondrial activities and the relevance to artemisinin resistance in these parasites. This point was discussed in the artemisinin resistance paragraph in the discussion.

- R1 pointed out image duplication, the reviewer is also concerned by some mishandling of data and statistics. For example, in Figure S7c the ESP15-AID strain + or – auxin show exactly the same data points.

Response: We thank the reviewer for these helpful comments. The previous figure was presented as a mean \pm SEM, which showed all the technical replicate results. In the new version of the analysis, we have re-exported the figure showing means \pm SD and which contains only the biological replicates (2 independent experiments).

More concerning is the fact that the authors considered technical replicates (i. e. triplicates within the same biological replicate) as biological replicates. In all the graphs the data points are shown together, this could seriously skew the statistics as technical replicates are linked while biological replicates are not.

Response: We thank the reviewer for these helpful comments and suggestions. In the previous version of the manuscript, we presented all the data analysis with mean \pm SEM, which showed up all the technical replicate results. We agree that technical points in the figures impair the statistical analysis and appearance of the figures. In the current version, we have re-analyzed the data and redone the figures, presenting only the biological replicates (2-3 independent experiments). We thank the reviewer for pointing out this problem.

For the quantification of the size of plaque assays, the authors only measured 9 plaques as stated in the legends of Figure 2f, while more than 100 plaques could have been scored. In figure 3h, data points are not shown in the graph.

Response: We thank the reviewer for these helpful comments and suggestions. We have re-analyzed the plaques and scored 18 plaques in each of the independent experiments, as presented in our previous study (pmid: 29269729). We have then

scored 54 plaques in total from three independent experiments for the TIR1 line, and other parasite lines. Considering that the sample size (plaque numbers) in the induced parasites is small, we consider that the present sample size is sufficient for the statistical analysis. We have re-analyzed the data in Fig. 3h to produce a new figure for the biological replicates. We thank the reviewer for these important suggestions to improve the statistical analysis.

Additional minor points:

- The author state that “Kelch13 is essential for the growth of *Toxoplasma* in mice” which they did not measure. They only measured strain virulence but not their ability to propagate in mice. The reviewer would suggest changing the title.
- The expression “extracellularly unstable” is unclear at best. It would be better to describe it as a “loss of fitness of extracellular parasites”.

Response: We thank the reviewer very much for these suggestions. We agree that the mouse infection measured virulence of the AID lines, not the parasite growth *in vivo*. We have thus revised the text, accordingly. We have also accepted the reviewer’s suggestions on the description of extracellular parasites, by changing it to loss of fitness of extracellular parasites in the manuscript text.

Reviewer #2 (Remarks to the Author):

Quite often sentences are hard to understand and it is sometimes guesswork what is meant. This makes me wonder whether the two native English speakers of the author’s list (GH, LDS) have read the manuscript thoroughly enough. GH’s contribution to the ms is not mentioned at all in the author contributions section!

Please clarify contributions, and work extensively on the text.

For the most part ok, but there are still numerous typos, omissions, etc, in particular in the newly added text (one of many: “Resources and reagents may to be directed to the corresponding author”)!

Response: We thank the reviewer very much for these helpful comments. We have carefully read the manuscript word-for-word for another five times and have corrected

the problems in the text. We have corrected the author's contributions.

“Generation of *T. gondii* Lines”: this chapter is hard to follow only by the information given in the tables, given the numerous different lines and constructs. At least the basic backbone of the non-commercial vectors should be provided as graphs, together with the insertion points of amplicons in the gene-of-interest. Moreover, not a single confirmative PCR for all the genomic insertions or KD is shown. While this might be considered ok for epitope or AID tagging, it does not exclude erroneous insertion at unintended loci. Please provide those PCRs at least for the most important parasite lines, and PCR primer locations added to the basic maps asked for above.

Response R1: We thank the reviewer very much for the comments. In this revised version, we have produced an illustration of the construction of the AID lines (Supplementary Fig. 3a-b), and made substantial revisions to the text on the methods describing the generation of generic plasmids, amplicons and lines. We generated the tagging plasmids from published and addgene-stored plasmids that we had previously submitted. The revised version contains the details of the addgene resources and descriptions of the steps in generating the constructs. Furthermore, the generation of the fusion lines exactly followed the methods described in our previous paper where more detailed information was supplied. We have repeated the diagnostic PCRs and included the full PCR results in the Supplementary Fig 3.

For the most part ok, although the number of stars (*, indicating “unspecific bands” in the endo PCRs) is surprising and asks for an explanation.

Response R2: We thank for the reviewer's concern on the PCR gels. We have thus made our observation and explanation in the Figure Legend. Here is the text.

The stars (*) indicated the nonspecific bands for the endoPCR products. We observed that the endoPCR products for EPS15-AID (3' tagging) in the single AID (EPS15-AID), dKD and tKD appeared to be different. It has to be noted that the EPS15-AID line served as the parental line for the generation of the double AID (dKD) line and the triple AID (tKD) line. In addition, the nonspecific bands in the endoPCR for AID-PPG1 (5' tagging) in the single AID line (AID-PPG1) and the tKD line looked different as well. Compared to the PCR with the TIR1 samples that contain specific priming sequences, the endoPCR in the AID samples seemed to readily produce non-specific products and even differently produced non-specific

products using different amounts of DNA in the sample. This is the likely the reason for why we generally see more than one non-specific band. The diagnostic PCR clearly detected the specific PCR products (integ PCR) from the AID integration in the AID parasites, but not the integPCR products in the parental line TIR1. We thus conclude that the AID lines were clean clones that contained a specific integration of AID at the targeted genes of interest used in this study.

In addition, the primers for the endoPCR could still amplify a long PCR product that contains the integrated heterologous DNA fragment (the AID-resistant cassettes with sizes > 4.0 kbp). However, the reaction was performed with a general TAQ DNA polymerase with a short extension time. This won't allow the PCR to make the long PCR products, but might produce different lengths of DNA products, as we suspected. This would thus produce complex fragments of DNA in different reactions.

We hope that this allows the diagnostic PCR to be better understood. We have added the above information in the Legend.

Fig 4: the quenching temperature in the method parts is conflicting the quenching temp mentioned in the Fig legend (-40° / -20°C). Several metabolites such as GABA, lysine, fumaric acid and glutamate should be detectable on both MS-platforms. How did the authors integrate these results? The fold-regulation data shown should not cherry-pick from either one experiment but integrate data from both experiments.

Response R1: We thank the reviewer for pointing out these problems. We have corrected the mistakes in the legend and methods. In the current version, we have provided results of LC and GC from both independent experiments in the main figure. As the two independent experiments were sampled at different temperatures (room temperature in Experiment 1 vs 4°C in Experiment 2), the outcomes of the metabolomics varied between experiments. We consider that the variations between experiments are mainly due to the sampling temperatures. In the former version, we considered it important not to waste resources on the measurements and included the outcomes of Experiment 1. However, we are not sure if it is necessary to present the data from experiment 1, or it is OK to present only the data from experiment 2? Though the outcomes of independent experiments varied, they were similar on most of the metabolites. We would be very interested to hear the suggestions from the reviewer.

It lies in the nature of experimental sciences that each experiment has to be performed multiple times, but at least twice, to distinguish robust from spurious data. In this respect it is very unfortunate that the two experiments differ by this drastic temperature difference during sampling. However, what is “N= “ referring to in the legend to Fig 4b (“Metabolic enrichment with differential metabolites identified in untargeted metabolomics from Experiment 1 (N=5) and Experiment 2 (N=3) (including LCMS and GCMS) for TIR1, AID-K13 and tKD after 36 hours of auxin induction.”)? Different numbers of cultures treated at different days but measured at one day?

Response R2: We thank the reviewer very much for their comments on metabolomics analysis, and we have accordingly revised the text and figure legend. “N” refers to the sample numbers for each of the parasite lines. Three of the lines (i.e. TIR1, AID-K13 and tKD) were grown and harvested in parallel, which were repeated five times on different days in experiment 1 (N=5). The samples were then measured in parallel by the LCMS and GCMS. We therefore considered the measurement as experiment 1. After finishing experiment 1, we prepared another round of samples, which were sampled in cold PBS. We repeated the sampling three times (N=3) on different days for three of the lines, and then carried out another biological replicate of measurement by LCMS and GCMS. We described the second batch of measurement as experiment 2. We thus reasoned that apparent differences between experiment 1 and experiment 2 are likely to have resulted from the sampling temperatures and the batch measurements.

We thank very much for the reviewer’s comments, and we have revised the description on the metabolomics in the Results and Figure Legends in the current version.

Also, what do you mean by “The empty boxes indicated the metabolites were not identified in experiment 1.”? There are empty green boxes and empty red boxes (as opposed to filled red boxes), but both, the green and red boxes contain data points. Please clarify. Depending on the answers it might be ok to present the two experiments as such (b-d).

Response R2: We thank the reviewer’s comments on the metabolomics analysis.

Experiment 2 obviously detected more metabolites, while experiment 1 had fewer metabolites identified. Therefore, some of the metabolites showed up with green and red columns (denotes metabolites identified in the parasite lines) in the boxes (metabolites identified by LC or GC) in experiment 2; however, the same metabolites were not detected in experiment 1, thus leaving the boxes empty (denotes metabolites not identified by LC or GC).

We thank the reviewer very much for pointing out the description problem and we have revised it in the Legend as follows. "The (empty or filled) color columns denote the metabolites identified in the parasite lines both in EXP1 and in EXP 2, while the empty boxes (without the color columns) in EXP 1 indicate the metabolites that were not identified by LC or GC."

Why is the triple kd less affected than the K13 kd in a number of metabolites (e.g. malate, glutamate, pyruvate, myristate and palmitelaidic acid)? Please comment.

Response: We considered that it might be due to depletion of additional proteins in tKD (EPS15 and PPG1), and they may have other unknown functions. This was what we can think of.

Given the problems with the metabolomics data (see above; Fig 4) I need to see those answers before I can agree.

Response: We considered that it is likely to be caused by additional depletion of proteins. However, we noticed that one of the readouts in the tKD line in experiment 2 was generally high for almost all of the metabolites detected in LC and in GC. This observation is supported by the results of metabolites identified in the same line (tKD) in experiment 1, where equal or lower readouts, compared to the readouts in AID-K13, were seen. We think that the higher level of metabolites in tKD in experiment 2 is caused by the experimental difference, but not by the strain difference.

We thank the reviewer very much for these comments and we have accordingly revised the text.

I recommend to make all images colorblind-friendly, in particular IFAs!

Response: We have done this following the suggestion, and revised the IFA images in Fig. 1a and 1b.

Why only those and not the others?

Response: We thank the reviewer very much for their suggestion. As that needs highly focused work to re-export, re-process and re-upload, we considered it best to retain the color images (most of them are in the supplementary figures), to avoid any image problems. Additionally, after four years of work on the project, the PhD student, who was responsible for the strain generation and IFA, has graduated and left the lab. I, as the senior author, will take on board this important recommendation and present images that are colorblind-friendly in our future work. Herein, I would like to kindly request understanding from the reviewer.

Reviewer comments, third round -

Reviewer #1 (Remarks to the Author):

The authors did alter their manuscript to address most of the reviewer's concerns.

I have just one remark left:

-In supp figure 7: an enlarged plastid is seen in the Auxin-treated parasites. The authors should mention it in the results and discuss it in the manuscript. Was that phenotype also seen using EM?

Reviewer #2 (Remarks to the Author):

The authors have responded satisfactorily to my comments on the first revision.

REVIEWERS' COMMENTS

Reviewer #1 (Remarks to the Author):

The authors did alter their manuscript to address most of the reviewer's concerns.

I have just one remark left:

-In supp figure 7: an enlarged plastid is seen in the Auxin-treated parasites. The authors should mention it in the results and discuss it in the manuscript. Was that phenotype also seen using EM?

Response: We thank the reviewer for his/her comments on the IFA observation. We have added the information to the Results section and made an appropriate comment to the observation. However, the Results section deals with the micropore from the angle of TEM and SEM on the structural changes. It would be better to just focus on these contents to clearly describe what we observed. Extending this part to the plastid would make the description complicated. Meanwhile, the enlarged plastid observed in the IFA won't be easily recognized on TEM. Instead, this would require a quantification and size measurement of the plastid for a final conclusion. So we would leave this part in a future study.

Reviewer #2 (Remarks to the Author):

The authors have responded satisfactorily to my comments on the first revision.

Response: We greatly appreciate the reviewer's positive evaluation of our work and response.